# Experiences of living with leprosy: A systematic review and qualitative evidence synthesis

**Norana Abdul Rahman** [1,2]*, **Vaikunthan Rajaratnam**[3], **George L. Burchell**[4], **Ruth M. H. Peters**[2], **Marjolein B. M. Zweekhorst**[2]

1 CRE, Perdana University, Kuala Lumpur, Malaysia, 2 Faculty of Science, Athena Institute, Vrije Universiteit Amsterdam, Netherlands, 3 Orthopaedic Surgery, Khoo Teck Puat Hospital, Yishun, Singapore, 4 Medical library, Vrije Universiteit Amsterdam, Netherlands

* norana.rahman@perdanauniversity.edu.my

**Data Availability Statement:** All relevant data are within the manuscript and its Supporting Information files.

## Abstract

### Objective

The objective of the review was to identify, appraise, and synthesise qualitative studies on the lived experience of individuals diagnosed with leprosy, the impact of the disease, and how they coped with the disease burden.

### Introduction

Leprosy is a chronic disease with long-term biopsychosocial impact and is a leading cause of preventable disabilities. It traps the individuals with leprosy in a vicious circle of disease, stigma, and poverty. The efforts to reduce stigma and discrimination and improve their quality of life have not kept pace with the success of the multidrug treatment.

### Inclusion criteria

This review considered published literature on the lived experience of individuals diagnosed with leprosy. There were no limitations on gender, background, or country. All qualitative or mixed-methods studies were accepted.

### Methods

The review followed the JBI meta-aggregation approach for qualitative systematic reviews. A structured literature search was undertaken using multiple electronic databases: PubMed, Embase, Web of Science, and CINAHL.

### Results

The search identified 723 publications, and there were 446 articles after deduplication. Forty-nine studies met the inclusion criteria. The final 173 findings were synthesised into ten categories and aggregated into four synthesised findings: biophysical impact, social impact, economic impact, and mental and emotional impact. These synthesised findings were consistent across the included studies from a patient's perspective. The way people coped with

**Funding:** The authors received no specific funding for this work.

**Competing interests:** The authors have declared that no competing interests exist.

leprosy depended on their interpretation of the disease and its treatment. It affected their help-seeking behaviour and their adherence to treatment and self-care. The review has identified a multi-domain effect on the affected individuals, which goes beyond the biological and physical effects, looking at the social issues, specific difficulties, emotions, and economic hardships.

## Conclusions

The researchers, health professionals, and policymakers could use the synthesised findings to address the concerns and needs of the leprosy-affected individuals and offer appropriate support to manage their lives.

## Systematic review registration number

PROSPERO Registration number: CRD42021243223

### Author summary

Leprosy is a chronic, granulomatous disease caused by *Mycobacterium leprae* and is a major cause of preventable disability. As a result of their disfigurements and deteriorating physical impairments, the leprosy-affected individuals experienced negative social attitudes, stigma, isolation, and discrimination, thus, impacting their lives and relationships with others. Due to the chronicity of leprosy, studying the lived experience of the affected individuals will allow us a deeper understanding of the effects of the disease, how they seek help and adhere to its treatment, and cope with the disease. We conducted a systematic literature review involving 49 articles, highlighting the four synthesised findings: biophysical impact, social impact, economic impact, and mental and emotional impact. These synthesised findings were not new, but they were consistent across the included studies from a patient's perspective. Our findings contributed to establishing the biopsychosocial and economic approach for understanding what leprosy-affected people experience by considering changes in their biophysical, sociocultural, psychological dynamics, and economics. It served as a guide for delivering care and treatment to these people. The information will be helpful to the researchers, health professionals, decision, and policymakers to plan and tailor their support for these individuals, their families, and the community.

## 1. Introduction

Leprosy is a chronic, granulomatous disease caused by *Mycobacterium leprae*. Multidrug therapy (MDT) for leprosy has been a huge global success. [1] However, if leprosy is left untreated or treatment is delayed, it can cause progressive and permanent damage to tissues and nerves, leading to skin sores, ulcerations, physical deformities, severe disfigurements, and disabilities. [2] It is now known that severe nerve complications leading to neuropathic pain can occur many years after completing treatment. [3] As a result, leprosy is a major cause of preventable disability [4], and an estimated 2–3 million people live with leprosy-related disabilities, suggesting a missed diagnosis or possible delays in seeking medical help. [5,6]

Leprosy still poses a public health challenge in the endemic countries, with India, Brazil, and Indonesia reporting 80% of the total number of new leprosy cases. [7] There is significant variation in the distribution of leprosy globally, and within some countries with low prevalence, there are local pockets of high endemicity. [8] The global prevalence of leprosy at the end of 2020 was 129,192, with a corresponding prevalence rate of 16.6 per million population. [9] There were 127,396 new cases detected worldwide, with a new case detection rate of 16.4 per million population, and these figures were much lower than in previous years, probably due to reduced leprosy prevention activities caused by the COVID-19 pandemic resulting in less detection and reporting of cases. [9]

People with leprosy can develop physical disabilities long before they are diagnosed, during, and after the completion of treatment. The diagnosis of new cases with grade 2 disabilities, these individuals have visible disabilities or deformities, is evidence of the missed disease or delayed treatment. The risk of disabilities is also high among defaulters. [10] dos Santos and colleagues [11] reported that despite being registered as treated and cured, 35% of leprosy-affected individuals in Cáceres-MT, Brazil, showed worsening signs of physical disability 15 years after their release from leprosy treatment. This finding indicates that follow-up should continue even after completing MDT. Additionally, many were lost to follow-up since they stopped being routinely evaluated by health professionals. [11] They were taught self-care and advised to return if their symptoms recur or worsen. [12] Some individuals did not receive timely treatment or follow-up because of their age, lack of knowledge, poor perception of symptoms, stigma, or inability to get to the health centres because of the distance and economic hardships. [5] Their leprosy status and worsening physical disabilities were not managed as they were excluded from active review. [11,13]

As a result of their disfigurements and deteriorating physical impairments, the leprosy-affected individuals experienced negative social attitudes, stigma, isolation, and discrimination, thus, impacting their lives and relationships with others. The efforts to reduce leprosy-related stigma have not kept pace with the success of the MDT. [14] Consequently, some leprosy-affected individuals might impose self-isolation on themselves. It could trigger a disequilibrium in family roles leading to families and community also excluding them from social activities. [15–19] Govindasamy and colleagues [20] carried out a multicentre, cross-sectional study in four leprosy-endemic states in India. They reported that thirty percent of people with leprosy who completed their MDT from the government and tertiary care hospitals in their states developed mental health issues. They suffered from negative psychological responses with feelings of guilt, low self-esteem, anxiety, and depression. [21,22] Studies have reported that people affected by leprosy have dropped out of school, were unemployed, or lost their jobs, which led to poverty and adversely impacted the household's income and quality of life. There was strong evidence of an association of poverty indicators with leprosy incidence. [23–25]

Health research is mainly influenced and dominated by positivist theories, but the social model sees experience and subjectivity as essential to the research process. [26] Health services mainly focus on curative medicine and the biophysical impact of disease but less on the psychosocial and economic aspects. Due to the chronicity of leprosy, studying the lived experience of the affected individuals will allow us a deeper understanding of the effects of the disease, how they seek help and adhere to its treatment, and cope with the disease. [27–29] The way they manage their lives depends on how they interpret their disease and attach meanings to their existence. [30] Their experience can inform the researchers, healthcare professionals, the decision-makers, and anyone interested in leprosy or who can make a difference to help improve their quality of life. [31]

Self-care is important for leprosy-affected individuals to encourage them to change their behaviour to adapt to the irreversible damage caused by the disease. [32] Self-care practice in leprosy is a daily activity that requires the individuals' active engagement to care for themselves, reduce the number of ulcers, prevent further deterioration, improve their physical wellbeing and increase self-confidence and self-esteem. Moreover, improved knowledge about leprosy will empower them to take appropriate and meaningful actions to help themselves and is key to a positive outcome. [32] Healthcare professionals can guide and support people with leprosy to self-care, improve their lives and access good healthcare and rehabilitation by employing effective communication strategies and providing psychological support and education-oriented counselling. [29,33] Most self-help centres run skills training workshops and microfinance schemes to help these people become independent. [4,34,35]

Coping with leprosy are the varying ways people manage stressful or traumatic situations to maintain their emotional wellbeing. Algorani and Gupta [36] identified four categories of coping: problem-focused, which is characterised by addressing the problems, such as talking to professionals to seek information, planning their activities, and getting expert help; emotion-focused to reduce negative thoughts and emotions by reframing and accepting the problems, using humour and religion; meaning-focused by using intellectual reasoning to manage the stressful situation; and social coping by requesting for support from their family, friends, and community. These coping styles can significantly impact the way people with leprosy understand their disease and how they manage their lives. [29,36]

There are many published studies exploring leprosy, its biophysical impact, and complications, including the stigma and discrimination faced by people with leprosy. However, to the best of our knowledge, there is no published qualitative evidence synthesis focused on the experiences of living with leprosy. Prior to the review, a preliminary search of PROSPERO, MEDLINE, the Cochrane Database of Systematic Reviews, and the *JBI Evidence Synthesis* was conducted. It did not identify any current or underway systematic reviews on the experiences of individuals living with leprosy.

Therefore, the primary purpose of this review is to identify, appraise and synthesise the available qualitative studies to examine the lived experience of the individuals afflicted with leprosy. We seek answers to the following research questions: i) What are the experiences of individuals living with leprosy? ii) What is the biopsychosocial impact of leprosy on these individuals? iii) What are their needs and the strategies they used to manage their disease? The synthesised findings will be helpful to the researchers, health professionals, decision-makers, and policymakers to tailor their support and guide the delivery of care and treatment to these individuals, their families, and the community.

## 2. Methods

This study presents a systematic review and qualitative evidence synthesis (QES) of the experiences of individuals living with leprosy to allow a greater understanding and provide rich interpretations of the experiences of the leprosy-affected individuals. [37] The systematic review was conducted following the Joanna Briggs Institute (JBI) methodology for systematic reviews of qualitative evidence. [38]

### 2.1 Search strategy

The search strategy aimed to locate the peer-reviewed publications for inclusion in the review. It was carried out with the guidance of GLB, a Medical Information Specialist, Vrije Universiteit, Amsterdam. An initial limited search of PubMed was undertaken on April 19, 2021, to identify articles on the topic, which yielded 164 articles.

A systematic search was then performed in the databases: PubMed, Embase.com, Clarivate Analytics/Web of Science Core Collection, and Cumulative Index to Nursing and Allied Health Literature (CINAHL). Various databases were used to ensure the results of the searches were comprehensive. The timeframe within the databases was from inception to April 28, 2021. GLB and NAR conducted the search for the articles. The search included keywords and free text terms for (synonyms of) 'Leprosy' combined with (synonyms of) 'quality of life' combined with (synonyms of) 'qualitative research'. The search strategy was also performed with the descriptors from the international vocabulary used in the health area, MeSH—Medical Subject Headings created by the National Library of Medicine for literature indexed on MED-LINE, combined through Boolean operators. Animal and child studies were excluded from the search. Studies published in any foreign language with English translation were included in the search. Non-English articles without full translation into English were excluded due to limited resources available for translational services.

A complete overview of the search terms per database can be found in the supplementary information (S1 Appendix).

The PICoS framework (Participants, Phenomenon of Interest, Context, and Types of studies), as shown in Table 1, was used to qualify the eligibility criteria. [39,40]

Studies that reported on children or adolescents were excluded as they were beyond the scope of the review. It did not consider studies in a foreign language, which did not have an English translation.

The only modification from the Systematic Review Protocol registered with Prospero during this search strategy was the exclusion of non-empirical articles. We only included peer-reviewed journal articles on the individuals' experiences of living with leprosy as there was sufficient published literature. We hoped to maximise the quality of the included studies. Other criteria followed the registered protocol.

## 2.2 Study selection

Following the search, all identified citations were collated and uploaded into EndNote 20 (Clarivate Analytics), and duplicates were removed. Following a pilot test, titles and abstracts of the articles were screened by two independent reviewers (NAR and RV) for assessment

**Table 1. PICoS Framework.**

| Participants | They are individuals of any gender and background, above 18 years old, diagnosed with leprosy. They may be new cases or those who have completed their treatment. |
|---|---|
| Phenomenon of Interest | The phenomenon of interest is the lived experience of the individuals living with leprosy. We specifically focused on the illustrations of these individuals with leprosy, looking at their problems and hardships, whether biophysical, psychosocial, economic, or other needs. |
| Context | The review considered studies of individuals with leprosy from any country, setting, or geographical location. |
| Types of Studies | This review considered qualitative and mixed-methods studies that clearly reported their qualitative component. Qualitative data was obtained using in-depth interviews, focus group discussions, and other qualitative methods such as observation and participatory methods. It included interpretive studies that drew on and described the experiences of the individuals living with leprosy, including, but not limited to, designs such as phenomenology, grounded theory, ethnography, action research, and feminist research. Mixed-methods studies with qualitative components and clearly reported findings were also considered. The studies included were original with aims and objectives or research questions clearly related to studying the experiences or their effects of living with leprosy. The studies' primary focus was on the experiences of living with leprosy rather than simply being a part of it. There was at least one relevant piece of information in the initial analysis to indicate its suitability for inclusion. |

against the inclusion criteria for the review. All potentially relevant studies were retrieved in full, and their citation details were imported into the JBI System for the Unified Management, Assessment, and Review of Information (JBI SUMARI) (JBI, Adelaide, Australia). [41]

The two reviewers (NAR and RV) independently evaluated in detail the full text of the selected articles against the PICoS framework. The reasons for the exclusion of these articles that did not meet the inclusion criteria were recorded and reported in S2 Appendix. Any disagreements that arose between the reviewers at each stage of the selection process were resolved through discussions. The results of the search and the study inclusion process were reported in full in the final systematic review and presented in a Preferred Reporting Items for Systematic Reviews and Meta-analyses (PRISMA) flow diagram (as shown under Results). [42] Forty-nine studies were included for this review [12,14,16,18,19,34,43–84] and are listed in S3 Appendix.

### 2.3 Assessment of methodological quality

Eligible studies were critically appraised by two independent reviewers for methodological quality using the standard JBI Critical Appraisal Checklist for Qualitative Research. [85] It involved a systematic assessment of each component for congruence and the influence imposed on the studies by the researchers. The two reviewers (NAR and RV) agreed that all studies would undergo data extraction and synthesis unless their appraisal scores dropped below 60/100. Such low scores would mean they are of poorer quality and did not meet a certain quality threshold for inclusion in the review. This decision was based on scores of five or more "No", "Unclear", or "N/A" responses to the JBI Checklist (S4 Appendix). Any disagreements that arose between the reviewers were resolved through further discussions. Authors of papers were contacted to request for missing or additional data for clarification. The results of the critical appraisal were reported in a table (S5 Appendix.)

### 2.4 Data extraction

The two independent reviewers (NAR and RV) extracted data from studies included in the review using the standardised JBI data extraction tool. [38] The data extracted included specific details about the populations, context, culture, geographical location, study methods, and the lived experience of individuals living with leprosy It further explored the themes, metaphors, and verbatim statements of the participants, which illustrated the impact of disease on their lives and how they coped with the disease burden across the different stages of the disease. The findings, and their illustrations, were extracted from the papers verbatim and assigned a level of credibility according to JBI: credible (C), not supported (N), or unequivocal (U). (S6 Appendix) The findings that were not supported were excluded from further stages of the review. Any disagreements that arose between the reviewers were resolved through further discussions. No other requests were made for additional data from the authors of the papers included in the review.

### 2.5 Data synthesis

The qualitative research findings were pooled using JBI SUMARI following the meta-aggregation approach. [38] The two independent reviewers (NAR and RV) initially reviewed all the findings. The review used the inductive approach to explore new phenomena based on answering specific research questions. This step involved the aggregation or synthesis of findings to generate a set of statements representing that aggregation by assembling the findings and categorising these findings based on similarities in meaning. The two reviewers agreed on the category titles. These categories were then subjected to synthesis to produce a single

comprehensive set of synthesised findings that was used as a basis for evidence-based practice. It is an iterative process involving comparison, refining, and reflection while synthesising the findings. Where textual pooling was not possible, the findings were presented in narrative form. The findings included in the synthesis mainly were credible findings.

## 2.6 Assessing confidence in the findings

The review used the ConQual [86] approach to grade the final synthesised findings to establish confidence in the output of qualitative research synthesis. The results are presented in a Summary of Findings table (S7 Appendix).

The Summary of Findings presents the overall dependability and credibility of the meta-aggregation process. Dependability was determined by the quality appraisal scores from questions 1 to 7 in the JBI critical appraisal checklist for qualitative research to determine whether study authors have met the specified criteria (S4 Appendix). Credibility was determined by the levels of credibility applied in the data extraction phase. Each synthesised finding from the review was presented, including the type of research informing it, the scores for dependability and credibility, and the overall ConQual score.

## 3. Results

### 3.1. Study inclusion

The electronic database search identified 723 records, of which 277 duplicate records were removed. A total of 446 titles and abstracts were screened for inclusion. A further 361 articles were removed because they did not fulfil the selection criteria, leaving 85 studies included for full-text screening, of which three articles could not be retrieved. Eighty-two studies were considered against the inclusion criteria, and another 33 were excluded for various reasons. The reasons for exclusion were that three articles were duplicates, ten did not meet the study design, nine had phenomena of interest that did not meet the inclusion criteria, and eleven articles were in another language, without English translation (S2 Appendix). A total of 49 studies were included in the review, and the process is reported in the PRISMA flow chart (Fig 1).

**3.1.1 Characteristics of included studies.** The 49 studies included in this meta-aggregation described the lived experience of approximately 1209 people with leprosy from 12 countries, with Brazil, Indonesia, and Nepal topping the list, as shown in Table 2.

Most of the included articles were published between 2010 to 2021 (n = 44) and the others were in 2009 (n = 1), 2006 (n = 3), and 2004 (n = 1). (Tables 3 and 4).

Some studies did not include full demographic information on their samples. Most studies had a combination of individuals with leprosy and others, including family members, community, and health professionals, but the review mainly focused on the individuals with leprosy. Purposeful sampling was the primary method used. There was a mix of male and female participants from different backgrounds with multibacillary or paucibacillary leprosy and varying grades of disabilities and reactions. Most studies did not specify what 'adults' meant, but the subjects were generally above 18 years old. (Table 4)

Most participants were outpatients, and the locations of data collection varied from hospitals or similar clinical sites to community centres and participants' homes. There were eight studies that were conducted in the leprosarium [14,34,49,52,58,71,73,75] and one in a correctional/prison facility [47]. All the studies used a qualitative approach or had a clearly reported qualitative component. There were nine studies based on the phenomenological approach [12,19,43,49,56,58,62,80,87], two were ethnographic studies [63,68], two used the grounded theory approach [55,78], and six were mixed methods [14,34,51,53,57,79], but only the

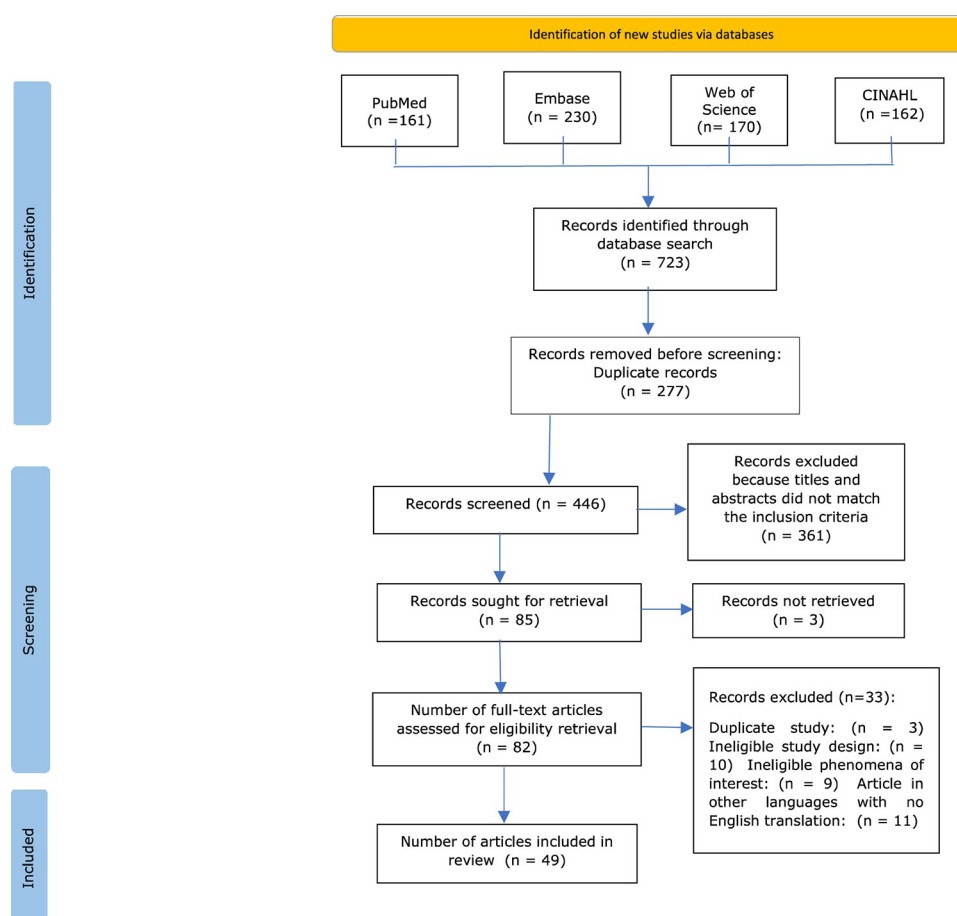

**Fig 1. PRISMA flow chart of literature search results, study selection, and inclusion process.**

**Table 2. Countries of origin of included studies.**

| Countries | Number of studies | Articles |
|---|---|---|
| Brazil | 18 | [44,45,47,48,54,59,60,63–66,72–74,76,81,82,87] |
| Indonesia | 10 | [12,16,18,51,56,62,67,70,80,88] |
| Nepal | 7 | [29,46,50,54,77,78,84,89] |
| Taiwan | 2 | [49,71] |
| Ghana | 2 | [52,75] |
| Nigeria | 2 | [34,53] |
| India | 2 | [68,69] |
| Iran | 1 | [43] |
| Korea | 1 | [58] |
| Surinam | 1 | [14] |
| Kiribati | 1 | [19] |
| Ethiopia | 1 | [83] |
| India and Indonesia | 1 | [79] |

**Table 3. Year of publication of included studies.**

| Year of publication | Number of studies | Articles |
|---|---|---|
| 2020 to 2021 | 13 | [19,56,58,59,62,65,69,72,78–80,83,89] |
| 2015 to 2019 | 19 | [12,14,18,34,49,50,52,54,60,61,66,67,73,75,76,82,84,87,88] |
| 2010 to 2014 | 12 | [16,43–45,47,48,53,64,68,70,74,81] |
| 2009 | 1 | [63] |
| 2006 | 3 | [46,71,77] |
| 2004 | 1 | [55] |

qualitative components were analysed. There was one study with a mixed disease population. [83] At least five of the studies were a part of larger studies. [16,18,51,61,67] The primary data collection methods were in-depth semi-structured interviews and focus group discussions. Most of the studies analysed their data using coding and thematic analysis. The table of characteristics of included studies is shown in Table 4.

## 3.2 Quality of included studies

The studies identified to be eligible for inclusion at this stage were critically appraised by the two independent reviewers (NAR and VR) using the JBI critical appraisal tool for qualitative studies (S4 Appendix). All the studies scored sixty or higher and underwent data extraction and synthesis. The 49 included studies were deemed moderate quality, as they had sufficient details that reported and demonstrated congruence between the research methodology, research questions, data reporting, and data analysis with sufficient illustrations of the participants and their voices. The most frequently absent detail in the included studies was the information on the influence of the researcher on the research (question 7, S5 Appendix); only five studies reported this item. [34,45,58,62,73] Seventeen studies [12,19,43,44,47,59,63,65,69–71,75–77,79,80,83] had no obvious statement locating the researcher culturally or theoretically. Ten studies [43,46,55,62,66,68,70,75,77,80] did not clearly state whether ethical approval had been granted by a relevant institutional committee. Results of the methodological appraisal of the quality of the included studies are shown in S5 Appendix as the Table of Critical appraisal results of eligible studies.

## 3.3 Review findings

The two reviewers deliberated over each included study. Following the JBI meta-aggregation process, findings were synthesised into categories and synthesised findings. There were 194 findings extracted from the 49 included studies to identify the experiences of the individuals living with leprosy. Each finding was accompanied by an illustration or a participant's voice and given a credibility level. Twenty-one findings were not supported as they lacked the participant's voice and were excluded from further stages of the review. The remaining 173 findings were assessed as credible findings and aggregated, based on similarities of meanings into ten categories, the descriptions of which were agreed upon by consensus between the two independent reviewers (NAR and RV). They were subsequently meta-aggregated into four synthesised findings, and they are namely: biophysical impact, social impact, economic impact, and mental and emotional impact. Table 5 (as shown below) illustrates the impact of leprosy on the lives of those afflicted by the disease.

Due to the number of findings retrieved from the 49 studies, only some examples will be used to demonstrate the findings and categories within each synthesized finding and these are shown in italics. S6 Appendix shows the complete list of study findings and their illustrations.

**Table 4. Characteristics of Included Studies.**

| Study | Methods for data collection and analysis | Country | Phenomena of interest | Setting/context/culture | Participant characteristics and sample size | Description of main results |
|---|---|---|---|---|---|---|
| Abedi H, Javadi A, Naji S. 2013. [43] | In-depth interviews and transcription, Colaizzi's 7-step phenomenological analysis | Iran | Health, family, and economic experiences of leprosy patients | Lorstan Province of Iran, community | Patients living with leprosy, sample size n = 10 | Three themes: 1. economic experiences: related to patients and family, 2. family experiences: related to the patients and family, 3. health experiences: physical experiences, psychological experiences, side-effects: feeling effects, emotional factors, and society's reactions. |
| Araújo de Souza I, Aparecido Ayres J, Meneguin S, Spagnolo RS. 2014. [44] | Recorded in-depth interviews, transcribed and reviewed, using the methodological strategy Collective Subject Discourse (CSD) | Brazil | From the perspective of complexity, to understand how people with Hansen's Disease perceive self-care. | The outpatient clinic of the Teaching Health Centre in Brazil | Outpatients leprosy patients, participants n = 15 | Four themes: 1. Experiencing Hansen's Disease 2. Perception of Self-care 3. Knowledge about drug therapy 4. Changes in lifestyle |
| Ayres JA, Paiva BSR, Duarte MTC, Berti HW. 2012. [90] | An exploratory study, qualitative approach- using semi-structured interviews. Interviews were transcribed, organised, central ideas and key expressions were extracted (Collective Subject Discourse). | Brazil | Repercussions of leprosy on the daily lives of patients. | Primary care unit, Faculty of Medicine, Brazil | Patients with leprosy n = 17 Multibacillary (MB) n = 11 paucibacillary (PB) n = 6 | Three themes: 1. The challenges of living with the symptoms and signs of leprosy 2. Relational life and leprosy 3. Daily life and leprosy |
| Carneiro da Silva RC, Araújo Vieira MC, Mistura C, Olinda de Souza Carvalho e Lira M, Sarmento SS. 2014 [47]. | An exploratory, qualitative approach using free word association (stimuli-evoked responses) and semi-structured interviews. Responses were recorded, transcribed, and analysed using content analysis. | Brazil | Perceptions of leprosy patients about stigma and prejudice faced in prisons. | Prison/correctional institution setting—one male and one female prison in Brazil | Inmates undergoing rehabilitation in correctional facility n = 7, men n = 5 women n = 2 MB n = 5 PB n = 2 | Free word association using the word 'leprosy' as stimulus: Treatment, Prejudice, Symptoms, and Feelings. Categories from Interviews: 1. Leprosy from the perspectives of patients and family relations, 2. Stigma of leprosy, 3. Impact of diagnosis and its feelings, 4. Denying leprosy and using divine faith to face the disease. |
| Carvalho e Silva Sales J, Ribeiro de Araújo MP, Cavalcante Coelho M, Lúcia Evangelista de Sousa Luz V, Araújo da Silva TC, José Guedes da Silva Júnior F. 2013 [48]. | A descriptive study using a qualitative approach, using semi-structured interviews. Interviews were recorded, transcribed, and analysed using content analysis. | Brazil | The perception of people with leprosy on their sexuality. | Reference centre for leprosy, Brazil | Participants with leprosy n = 10 men n = 6 women n = 4 | Two themes: 1. Sexuality is a synonym for sex in the leprosy patient's view. 2. The repercussions of leprosy in the sexuality of people with the disease. |
| Chen IJ, Cheng SP, Sheu SJ. 2017 [49]. | Qualitative approach using face-to-face in-depth interviews using semi-structured interview guide. The interviews were recorded, transcribed, and analysed using the Colaizzi method of thematic analysis. | Taiwan | The meaning of physical activity for elderly leprosy patients who had experienced isolation. | A leprosy sanatorium in Taiwan | Participants living in sanatorium n = 23 men n = 16 women n = 7 Average age: 73y Speaks Chinese or Taiwanese | Four themes (T) and nine categories (C) T1: Physical activity is a natural component of life - C1: A natural tendency, C2: Part of everyday routine. T2: Physical activity is beneficial to one's body and mind - C3: Health benefits, C4: Brings me joy, peace, and hope, C5: Empowers me. T3: Difficulty with physical activity is a degrading reminder of leprosy - C6: Brings out bitterness and frustrations from my past, C7: Exposes my inferiority T4: Physical activity is the acceptance of one's life circumstances - C8: Reminds one's coexistence with leprosy C9: Provides a field for one's cultivation |

*(Continued)*

**Table 4.** (*Continued*)

| Study | Methods for data collection and analysis | Country | Phenomena of interest | Setting/context/culture | Participant characteristics and sample size | Description of main results |
|---|---|---|---|---|---|---|
| Correia JC, Golay A, Lachat S, Singh SB, Manandhar V, Jha N, et al. 2019 [50]. | Qualitative study, semi-structured interviews (until saturation) based on the "5 dimensions of the educational diagnosis" framework. The interviews were recorded, transcribed, and analysed using thematic analysis, and the coded themes were organised according to the model. | Nepal | treatment. | Two leprosy referral centres: 1. A university hospital 2. A primary health care centre in Nepal | Patients diagnosed with leprosy n = 11, males n = 7 females n = 4, Healthcare workers n = 15 Speaks English or a Nepali dialect | Five dimensions: 1. Socio-professional dimension: low socio-economic status, unable to work, unemployed, and stopped schooling 2. Biomedical dimension—late treatment, disabilities, difficulty in accessing care 3. Cognitive dimension—alternate health beliefs, failure to recognise symptoms 4. Psycho-affective dimension—stigma and social exclusion, suffering stress and anxieties 5. Patient projects—unfamiliar with the concept. |
| da Silva Duarte LMCP, Albino Simpson C, dos Santos Silva TM, de Lima Moura IB, Ramos Isoldi DM. 2014 [81]. | A descriptive, exploratory, qualitative study using an identification questionnaire and semi-structured interviews were recorded and analysed using Bardin's content analysis. | Brazil | Self-care actions taken by leprosy patients | Dermatology outpatient sector of a University Hospital, Brazil | Patients with leprosy n = 14 Men n = 5 Women n = 9 current or former leprosy patients | Three themes: 1. Leprosy complications are known by people with leprosy. 2. Self-care actions practised by people with leprosy. 3. Possible contributions of a self-care group for people with leprosy. |
| da Silva Santos K, Magali Fortuna C, Fagundes Carvalho Gonçalves M, Matumoto S, Ribeiro Santana F, Marciano FM. 2015 [82]. | A qualitative study using in-person semi-structured interviews. The interviews were recorded and transcribed for analysis based on the Vigotskian approach | Brazil | The meaning of leprosy for people treated during the sulfonic (1944–1986) and multidrug therapy (MDT) (1986- present) periods. | At a MORHAN (Movement for reintegration of people affected by leprosy) centre, Brazil | MORHAN members leprosy patients n = 8, i) treated during sulfonic period n = 4, ii) treated with MDT n = 4 | 3 themes: First meaning core: 1. Spots on the body—something is out of order. 2. Leprosy or hanseniasis? 3. Leprosy from the inclusion at MORHAN |
| Da Silva MCD, Paz EPA. 2019 [87]. | Semi-structured interview, end-point—when data saturation was achieved. Interviews were recorded, transcribed, and analysed using the hermeneutic circle based on Gadamer. | Brazil | Meaning of experiences of people affected by leprosy during treatment in the health services | A designated room at a health facility in Brazil | Participants n = 21, men n = 13, women n = 8, PB n = 9, MB n = 12, cured n = 9, leprosy reactions n = 8, physical impairments n = 5 | Three categories 1: The disease makes life unsettling and painful, 2: Turnover of professionals causes insecurity during the evolution of the disease and treatment, 3: To protect themselves from prejudice, people with leprosy adopt attitudes to reduce social tensions. |
| Dadun, Peters R, Lusli M, Miranda-Galarza B, van Brakel W, Zweekhorst M, et al. 2016 [51]. | A qualitative approach using in-depth interviews and focus group discussions. Content and thematic analysis were done. | Indonesia | The barriers that exist and how they can be dealt with to implement a socio-economic development intervention. | Cirebon, Indonesia—setting unclear | Persons affected by leprosy n = 53 women n = 34 men n = 19, others: healthcare providers and key persons in the community | 1. Socio-economic consequences of leprosy 2. Barriers in the health system 3. Barriers related to knowledge, beliefs, and attitudes in society 4. Barriers related to emotional and physical consequences of leprosy |
| Dako-Gyeke M, Asampong E, Oduro R. 2017 [52]. | A qualitative research using in-depth interviews. Interviews were recorded, transcribed and analysed using thematic analysis. | Ghana | Experiences of people affected by leprosy—their perceptions about stigmatisation and discrimination | A leprosarium in Southern Ghana | People affected by leprosy n = 26 males n = 16, females n = 10, living in the leprosarium | 1. Lack of knowledge about leprosy and stigma 2. Barriers to accessing healthcare facilities 3. Barriers to accessing employment opportunities 4. Factors that affect their reintegration into their respective communities |
| Ebenso B, Ayuba M. 2010 [53]. | Mixed methods- Participation scale and a qualitative component using semi-structured interviews of leprosy patients, FGDs, and key informant interviews. Qualitative interviews and group discussions were recorded, transcribed verbatim, and analysed using thematic analysis. | Northern Nigeria | Perceptions of people with leprosy on the effect of socio-economic rehabilitation (SER) on leprosy-related stigma | Northern Nigeria, setting unclear | Total participants n = 65 leprosy patients affected by leprosy involved in socio-economic rehabilitation n = 20. Others included family members/neighbours of the 20 patients and community members (key informants). | Six themes 1. Prejudicial attitudes 2. Independence and income generation 3. Access to public institutions 4. Desire for acceptance 5. Dignity 6. Components of SER that stimulate social interaction. |
| Gonçalves M, Prado M, Silva SSD, Santos KDS, Araujo PN, Fortuna CM. 2018 [54]. | An exploratory, qualitative study semi-structured interviews and field diaries—to record observations of behaviour during the interviews, personal impressions, etc. All interviews audio-recorded, transcribed verbatim and analysed using thematic content analysis. | Brazil | The interference of leprosy in the lives of women in relation to work and activities of daily living. | Health service in an urban municipality of Sao Paulo, or interviews also done at the home of the participants | Women with leprosy doing a variety of work activities n = 10 | Three categories: 1. The pains of leprosy 2. Changes with disease and adaptations in work and activities 3. Being a woman with leprosy |

(*Continued*)

**Table 4.** (Continued)

| Study | Methods for data collection and analysis | Country | Phenomena of interest | Setting/context/culture | Participant characteristics and sample size | Description of main results |
|---|---|---|---|---|---|---|
| Heijnders ML. 2004 [29]. | Qualitative study using in-depth interviews. The interviews were tape-recorded, translated and transcribed, and analysed using a grounded theory approach and the pattern matching methodology. | Eastern Nepal | The different coping strategies employed by leprosy patients to manage stigma. | Interviewed in their homes, Nepal | Participants interviewed n = 76 Those who discontinued treatment n = 29, Those who had completed and released from treatment n = 47 | 1. Strategies of concealment caused by expected stigma, 2. Strategies employed in managing experienced stigma 3. Concealment cycle—mutual concealment, wait and see approach, 4. The importance of triggers-impacts on coping strategies and level of stigmatisation, 5. The importance of social differentiation in stigma—depends on interactions between people and their positions within the hierarchies |
| Jatimi A, Yusuf A, Andayani SRD. 2020 [56]. | A descriptive, qualitative study with a phenomenological approach using in-depth semi-structured interviews. All interviews were recorded and analysed by thematic analysis (manually) and member check was carried out. Field notes were also kept. | East Java, Indonesia | The experience of leprosy sufferers with disabilities shows resilience. | Conducted in the participants' homes, East Java | Leprosy patients n = 11, mostly women, age 25-34y, mostly grade 1 disability, 6-month treatment programme | Five themes: 1. Self-stigma—labelling, discrimination 2. Psychosocial problems—anxiety, withdrawal, self-concept disorders 3. Active coping—utilising social support, spirituality, distraction techniques 4. Positive adaptations—active work, social interaction, adjustment of physical conditions, self-respect 5. Characteristics of resilient individuals—responding positively to undesirable conditions, being more productive, helping others |
| Jha K, Choudhary RK, Shrestha M, Sah A. 2020 [57]. | Qualitative and quantitative methods were used—FGDs, In-depth interviews, and the general self-efficacy scale. Quantitative data—statistical analysis Qualitative data—was coded manually and thematic analysis was done. | Nepal | Women's empowerment in mixed self-help groups (SHGs). | Lalgadh leprosy hospital and services centre, Dhanusha District, Nepal | Women with leprosy from the SHGs n = 40, non-SHGs n = 20, key informants n = 10, and women leaders of Dhanusha SHGs n = 8. | Women from SHGs 1. Good knowledge about their rights 2. Higher participation in social programmes—opportunities in the planning and decision-making tasks 3. More employment through their engagement in income-generating activities and saving schemes connected to SHGs Barriers faced by women in SHGs: 1. male dominance 2. lack of assets and resources 3. social and cultural barriers 4. lack of capacity 5. domestic violence 6. defamation |
| Jung HG, Yang YK. 2020 [58]. | A phenomenological research methodology using in-depth interviews to the point of saturation. All interviews were audio-recorded, transcribed, analysed using Colaizzi's analysis, and the results were validated. Interview logs were also kept and used for analysis. | Korea | The life experiences of female leprosy patients residing in a leprosy settlement village in South Korea | A leprosy settlement village in South Korea | Female leprosy patients n = 11, mean age 75y, lived in the leprosy settlement | Four theme clusters (TC) and nine themes (T) TC1. inescapable shackles T1. changing body and living in hiding T2. barriers too high to overcome TC2. Suffered as if being in prison T3. Left the family and entered a confinement facility to live T4. hellish life at the confinement facility TC3. In no position to be a woman or a mother T5. Lost feminity and motherhood T6. Crow in crow T7. deformed body TC4. Another hometown T8. Backroad of life stained with suffering T9. self-healing at a second sanctuary |

*(Continued)*

**Table 4.** (Continued)

| Study | Methods for data collection and analysis | Country | Phenomena of interest | Setting/context/culture | Participant characteristics and sample size | Description of main results |
|---|---|---|---|---|---|---|
| Khanna D, de Wildt G, de Souza Duarte Filho LAM, Bajaj M, Lai JF, Gardiner E, et al. 2021 [59]. | A qualitative study using maximum variation and snowball sampling and one-to-one in-depth semi-structured interviews. Data were analysed using a constant comparative method which determined the point of saturation—thematic analysis and deviant case analysis. N-Vivo was used to code the data. | Brazil | The experiences, perceptions, and beliefs of leprosy patients and their carers influencing treatment outcomes. | Leprosy patients residing in Petrolina, Pernambuco, NE Brazil | Leprosy patients who are being retreated due to poor treatment outcomes n = 14. Carers of these patients n = 13. | Four patient themes 1. Personal factors a. Knowledge and information quality b. Health beliefs c. Psychological impact and character 2. External factors a. Socioeconomic factors b. Structural factors c. Support factors 3. Clinical factors a. Treatment and side-effects b. Experiences of diagnosis 4. the HCP-patient-carer relationship |
| Sottie CA, Darkey J. 2019 [75]. | Qualitative approach using individual interviews and FGDs. The framework approach was used to analyse the data—identifying themes and concepts and detecting patterns. | Ghana | The experiences of leprosy patients who have been cured in living with stigma and discrimination from society. | Cured lepers' village (CLV) at Ho in Ghana | Participants n = 20, people cured of leprosy, residents of CLV n = 15, family members n = 2, an officer at Ho Polyclinic, a caretaker of the village and an official of the Ho Social Welfare Department | 4 Themes: 1. Length of stay in the CLV 2. Isolation, abandonment, and neglect 3. Verbal abuse and ridicule 4. Self-stigma and shame |
| Yusuf A, Aditya RS, Yunitasari E, Aziz AN, Solikhah FK. 2020 [80]. | A qualitative phenomenological study based on an interview protocol All interviews were audio-recorded, transcribed and thematic analysis was done—first by the researchers independently and then NVivo and a final review of all transcripts, themes and patterns. | Indonesia | The psychosocial problems experienced by persons affected by leprosy | Conducted in the working area of East Java Province | Participants with leprosy n = 16 | Three themes: 1. Anxiety: symptoms, intensity 2. Pulling away: barriers to social interaction 3. Impaired self-concept: body image, pride |
| Van Netten WJ, Van Dorst MMAR, Waltz MM, Pandey BD, Aley D, Choudhary R, et al. 2021 [78]. | A Qualitative approach, purposive maximum variation sampling, semi-structured interviews, data saturation. The interviews were audio-recorded, translated into English, and transcribed verbatim. The transcripts were coded based on themes from the framework using the analysis programme ATLAS-ti. A framework method was used to analyse the data. It also had the Grounded Theory Approach characteristics using both deductive and inductive approaches. | Nepal | The mental wellbeing of people affected by leprosy | Leprosy patients who are part of the SHGs run by the LLHSC (Lalgadh Hospital). | Leprosy patients n = 14 men n = 4, women n = 10, disability grade 1 n = 6, grade 2 n = 8, experts in leprosy n = 2. | 1. People's feelings and experiences regarding leprosy: a) mental wellbeing b) stigma c) severity of disability d) disease concepts 2.Social and daily life factors: family and community work 3. Cultural factors: a) knowledge about leprosy b) gender c) religion 4. Self-help groups and counselling |
| van Haaren MAC, Reyme M, Lawrence M, Menke J, Kaptein AA. 2017 [14]. | Mixed-methods study: Questionnaire -B-IPQ and semi-structured interviews to assess the cognitive and emotional representations of illness. The common sense model was used. Quantitative data: SPSS—descriptive statistics and frequencies, independent t-tests Qualitative: interviews were recorded, transcribed verbatim and quotations from the interviews were used to contextualise the illness perceptions. | Surinam | Leprosy cured participants' illness perceptions and their resulting behaviour. | At a leprosarium run by the Esther foundation in Surinam | Leprosy-cured patients (diverse ethnicity) n = 13 Healthcare professionals n = 11 | 1. Illness perceptions: expected effects of illness 2. Perceived causes: family curse, food, and heredity 3. Concealment: coping mechanism |
| van 't Noordende AT, van Brakel WH, Banstola N, Dhakal KP. 2016 [84]. | Cross-sectional data and semi-structured interviews using an interview guide consisting of 4 themes: sense of self, marital relationship, knowledge, awareness of sexual and reproductive health, and sexual relationship. All the interviews were audio-recorded, translated, and transcribed in English and analysed using open coding and content analysis in MAXQDA. | Nepal | The experiences of women with leprosy regarding marital life and sexuality. | Interviews were carried out in participants' home, or a private, safe space near their homes. | Women affected by leprosy n = 10, grade 2 disability n = 4 grade 1 n = 6, 10 with visible physical impairment and ten able-bodied women. | Main themes: 1. Marriage, sexual relationships and sex education 2. Factors affecting the marital relationship of women 3. Factors affecting the sexual relationship of women |

*(Continued)*

**Table 4.** (Continued)

| Study | Methods for data collection and analysis | Country | Phenomena of interest | Setting/context/culture | Participant characteristics and sample size | Description of main results |
|---|---|---|---|---|---|---|
| Try L. 2006 [77]. | A qualitative study using semi-structured interviews. All interviews were recorded, transcribed, and analysed using thematic content analysis. | Nepal | The experiences of leprosy-affected people on marriage and stigma due to leprosy. | Lalgadh leprosy services centre in SE Nepal. | Leprosy patients n = 19, women n = 10, men n = 9 | 1. Stigma: The perceptions and behaviour—perceptions and beliefs about leprosy, stigmatised behaviour 2. Marriage—The stigma of leprosy impacts on the marriage of leprosy-affected individuals and the marriage prospects of relatives (courtesy stigma) 3. Attitude and expectations: the ways a community stigmatised leprosy-affected individuals and how each individual perceives the stigma |
| Thompson L, Iotebab N, Chambers S. 2020 [19]. | A qualitative design using an interpretive phenomenology: semi-structured interviews. Interviews were audio-recorded, transcribed and translated into English. The data were analysed thematically using Braun and Clarke's six step guide. | Kiribati, Pacific Islands | The lived experience of people living with leprosy in Kiribati | Community clinic, Tarawa Island, Kiribati | Leprosy patients n = 7, males n = 3, and females n = 4 | Two main themes 1. Recognising leprosy 2. Stigma |
| Susanto T, Dewi EI, Rahmawati I. 2017 [12]. | A descriptive, qualitative study using a phenomenological approach —FGDs were held using semi-structured interviews The discussions were transcribed verbatim and analysed using Colaizzi's seven steps | Indonesia | The experiences of leprosy patients who participated in self-care groups in the community. | Two self-care groups (in the community) in Jember, Indonesia | Leprosy patients n = 17 males n = 10 and women n = 7, | 5 categories: 1. Self-perceived condition a) understanding of the diseases b) self-image related to their disease 2. Adherence to treatment a) Lack of confidence in the treatment provided by PHCs b) Understanding of MDT short-term and long-term therapy regimens c) Efforts to reduce side effects of treatment 3. Ability to do self-care a) Ability to do meet basic human needs b) Control of the living environment c) The use of personal protective equipment d) Skin and wound care and prevention of disability e) Participation in the self-care group 4. The kind of help and services received a) Those based on ancestral cultural/religious heritage b) Traditional and alternative medicine c) Modern services from health workers 5. Acceptance and support for leprosy patients a) Family support during treatment b). Public social acceptance towards leprosy clients c) Provision of adequate information and health services from public health centres d) Return to work and acceptance at work after recovery |

(*Continued*)

**Table 4.** (Continued)

| Study | Methods for data collection and analysis | Country | Phenomena of interest | Setting/context/culture | Participant characteristics and sample size | Description of main results |
|---|---|---|---|---|---|---|
| Steremberg Pires D'Azevedo S, Nunes de Freitas E, do Nascimento LO, dos Santos DCM, Delmondes do Nascimento R. 2018 [76]. | A qualitative, descriptive study, random sampling using semi-structured interviews, until saturation was achieved. The interviews were analysed using thematic analysis | Brazil | The perceptions of leprosy patients about self-care support groups (SSG) that reduce physical disabilities through preventative measures. | The study was carried out at a General hospital in Recife, Brazil. | Leprosy patients n = 11, men = 8, women n = 3. They were mostly MB. | Two categories: 1. importance of self-care for the prevention of physical and psychological disabilities—provided them with social interactions and the ability to recognise the need to adopt self-care. 2. Contributions of the SSG in coping with the difficulties and limitations of the people affected by leprosy |
| Silva CA, Albuquerque VL, Antunes MF. 2014 [74]. | A qualitative study using semi-structured interviews and informal conversations. The interviews were recorded, transcribed, and analysed using thematic analysis. | Brazil | Patients' perceptions of leprosy and the stigma they experienced in the family, in their social life, and in their professional environments | The research took place at the National Reference Centre for Dermatology in Fortaleza, NE Brazil | Leprosy patients n = 20, Women n = 10, men n = 10 | Categories: 1. Changes that occurred in the family—reactions of religiosity, prejudice and aversion, guilt, fear, and escape 2. Neighbours' and co-workers' attitude—Not informing work colleagues, hiding from neighbours and co-workers for fear of dismissal and prejudice 3. Consequences of leprosy in patients' social lives—low self-esteem, nervousness, preoccupation, and tendency towards isolation. |
| Sillo S, Lomax C, De Wildt G, Fonseca MD, Galan NGD, Prado RBR. 2016 [73]. | A qualitative study, semi-structured, face-to-face interviews. They were audiotaped and transcribed into Brazilian Portuguese and transcripts were translated into English. Data were managed using NVivo10 software—analytic memos were kept, codes were sorted into categories and sub-categories and thematic analysis was carried out. A reflexive diary was also kept to improve trustworthiness. | Brazil | The experiences of current and ex-leprosy patients on the impact of stigma on their lives in Brazilian society | The study was conducted at a former leprosarium ILSL, in Sao Paulo, Brazil, which is now a leprosy research and referral centre | Leprosy patients n = 27, Males n = 16, females n = 11 MB leprosy n = 21 and PB n = 6. 23 were whites, 2 blacks and 2 browns. | Themes 1. Changing attitudes towards leprosy patients in Brazil a) Fear of disease due to misconceptions b) Abolishment of mandatory isolation, MDT, contact tracing, vaccination, using media in public health campaigns 2. Discrimination experienced by leprosy patients a) Discrimination by the government b) Discrimination by family members c) Discrimination in the workplace 3. Complications of disability a) Experienced stigma, perceived stigma, and self-stigma b) Caused inability to work and loss of social participation |
| Lima MCV, Barbosa FR, Santos D, Nascimento RDD, D'Azevedo SSP. 2018 [60]. | A descriptive, qualitative study using one-to-one semi-structured interviews. Interviews were recorded and analysed using content analysis. | Brazil | Self-care practices for face, hands, and feet executed by leprosy patients | Study conducted in healthcare units in Recife, Pernambuco, Brazil | Leprosy patients n = 24, mostly males with varying degrees of disability | Two thematic categories: 1. Knowledge and execution of self-care practices in Hansen's disease 2. Singularities and challenges to self-care in leprosy. |
| Shyam-Sundar V, De Wildt G, Virmond MCL, Kyte D, Galan N, Prado, et al. 2021 [72]. | A qualitative study, in-depth, face-to-face semi-structured interviews. Purposive sampling with maximum variation of characteristics. All interviews were audio-recorded, transcribed verbatim, and translated into English. Data were analysed in English—they were coded, grouped into categories using mind-maps and the sub-categories were used to identify four emerging themes. All interview transcripts were managed using NVivo and Braun and Clarke's 6 step process was applied to thematic analysis. Notes of non-verbal communication eg. body language were made throughout all the interviews. | Brazil | The perceptions of leprosy-affected persons on the impact of race on their access to leprosy treatment and care as well as their experiences of leprosy-related stigma and discrimination in society. | A private side room at a public funded leprosy research and tertiary referral centre (ILSL) in Sao Paulo, Brazil. | Patients with leprosy n = 20, inpatients at the ILSL n = 2, outpatients n = 18. blacks n = 5, whites n = 8, browns n = 6, and indigenous n = 1. | Themes: 1. Racism is part of Brazilian culture a) Blacks are the most discriminated race in Brazil b) No direct role of race in leprosy treatment 2. Difficulties associated with leprosy diagnosis a) Stigma and discrimination b) Lack of knowledge 3. Barriers to accessing care a) Lack of adequate treatment centres b) Stigma from healthcare professionals 4. Lack of health education a) Education through schools b) Improve health education strategies |

(*Continued*)

**Table 4.** (Continued)

| Study | Methods for data collection and analysis | Country | Phenomena of interest | Setting/context/culture | Participant characteristics and sample size | Description of main results |
|---|---|---|---|---|---|---|
| Shieh C, Wang HH, Lin CF. 2006 [71]. | Qualitative inquiry method using individual interviews and FGDs. The interviews were audiotaped, transcribed, and analysed in 2 phases using structural analysis (identified life stages) and holistic content analysis. | Taiwan | Stories told by women who had suffered leprosy from the beginning of the disease to the recovery | The interviews were carried out at the Lo-Sheng leprosarium, a Leprosy long term care centre (LLTCC), the community, in Taiwan | Women n = 21: 12 lived in the leprosarium, 5 lived in the LLTCC and 4 (former patients of LLTCC) in the community. | 1. Before being diagnosed a) Aware of being different from others in the early stage of the disease but did not know the cause of the problem. b) Symptoms would not go away after many attempts to get rid of them. 2. After being diagnosed a) Feeling of shame and being stigmatised b) Conflicting ideas about being admitted to institutions for treatment c) Motivated to receive treatment and wanted to be leprosy free 3. Living with leprosy a) Missing early attachment to own children b) Loving, protecting, and being proud of own offspring c) Striving for family, social, and religious support d) Being as useful and independent as possible 4. The Future a) Once leprosy, forever leprosy b) Rather stay in the leprosarium or the LLTCC than move out to the community c) Life is more difficult with pain, disabilities, and chronic diseases d) Welcome the final days to come |
| Schuller I, van Brakel WH, van der Vliet I, Beise K, Wardhani L, Silwana S, et al. 2010 [70]. | Mixed methods—Quantitative and qualitative studies Quantitative: Rapid disability appraisal toolkit Qualitative: In-depth interviews and FGDs | Sulawesi, Indonesia | The experiences of rural women and how they cope with disabilities, especially disabilities related to leprosy. | The research was conducted in two rural settings around Makassar in Sulawesi, Indonesia. | Five different locations: In each location—In-depth interviews: 1 woman with leprosy-related disability and another with other disabilities and key informants (2 community leaders, and 1 religious leader) FGDs: about 7 disabled women in each group | Main themes from the women with disabilities: 1. Work, 2. social activities, 3. acceptance by the community |
| Ramasamy S, Govindharaj P, Kumar A, Panneerselvam S. 2020 [69]. | A qualitative study using semi-structured interviews. The process of analysis was unclear. | India | The challenges regarding disclosure of disease among women affected by leprosy | Tertiary hospital at Champa, India | Women, mostly affected by MB leprosy n = 57 | Disclosure of disease: a) Fear of disease, b) positive experience after disclosure, c) violent reaction after disclosure, d) psychological issues, e) self-stigma, f) problems in practising self-care, g) stigmatization and family members, h) beliefs and myths attached to leprosy, i) experience of societal stigma, j) employment issues |
| Lusli M, Peters R, Bunders J, Irwanto I, Zweekhorst M. 2017 [61] | An exploratory qualitative study using in-depth interviews and FGDs. All interviews were recorded, transcribed, and translated into English. All data were divided into categories and analysed thematically. | Cirebon, Indonesia | The characteristics (low self-esteem, feeling guilty), perceptions of leprosy, experiences with the disease, and needs of the people affected by leprosy and those of the community regarding leprosy and stigma | Location: SARI office or at a hotel, Cirebon, Indonesia | 53 In-depth interviews— leprosy patients n = 44 and carers of children affected by leprosy n = 9. 38 participants in 5 FGDs | 1. health perceptions 2. feelings and emotions 3. human rights and discrimination The exploratory study influenced the counselling practice in 4 ways: 1. lack of knowledge about leprosy and its consequences 2. guilt and low self-esteem 3. violation of human rights 4. stigma manifests at many levels and in different ways Key issues identified during pilot How the pilot informed the counselling practice and model |

*(Continued)*

**Table 4.** (Continued)

| Study | Methods for data collection and analysis | Country | Phenomena of interest | Setting/context/culture | Participant characteristics and sample size | Description of main results |
|---|---|---|---|---|---|---|
| Lusli M, Zweekhorst MB, Miranda-Galarza B, Peters RM, Cummings S, Seda FS, et al. 2015 [18]. | A qualitative study using in-depth interviews (IDIs) and FGDs. All interviews were recorded, transcribed and analysed using thematic analysis. | Cirebon, Indonesia | Experiences of people affected by disabilities and leprosy | Office of SARI project/social office, Cirebon, Indonesia | IDIs: 14 participants—7 with leprosy and 7 with other disabilities, 3 FGDs: 13 (7 with leprosy, 6 other disabilities), 9 with leprosy, 9 with disabilities | 1. Stigma: 4 domains - a) emotions b) thoughts c) behaviour d) relationships Many similarities between leprosy patients and those with disabilities. Those with leprosy frame their problems in medical terms as patients suffering from disease even after being cured. Those with disabilities from a social perspective may have an abnormal body, but they are not sick. |
| Poestges H. 2011 [68]. | An ethnographic study uses semi-structured interviews, informal conversations, and participant and non-participant observations. The interviews were tape-recorded in Tamil and later transcribed and translated into English. Data were analysed using ethnographic methods | India | The perceptions of a leprosy colony from the perspectives of the colony members and the people (with/without leprosy) living in the adjacent neighbourhoods outside the colony | The interviews were conducted in and in-front of the participants' homes | Three groups: 12 people with leprosy and impairments and 10 of their relatives, 14 inhabitants of the adjacent neighbourhood with no leprosy, 11 leprosy-affected people living in the neighbourhood outside the colony. | Results: 1. Stigma in and around the leprosy colony 2. Narratives about the origin of the colony 3. Creating and maintaining community membership 4. The concept of a collective leprosy-affected body 5. Stigma as a means of fund-raising 6. Distinctive social values within the colony 7. Control over community membership 8. Leprosy versus colony member identity |
| Peters RM, Zweekhorst MB, van Brakel WH, Bunders JF, Irwanto. 2016 [67]. | Qualitative methods- 1. semi-structured interviews with the participants before and after the process 2. informal discussions with the participants during the process 3. participant observation with a focus on participants, the process and areas for improvement 4. photos and videos of the process 5. notes of the initial and evaluation meetings with the research assistants 6. written reflections by research assistants on challenges and opportunities. The interviews were recorded, transcribed verbatim or comprehensively summarised with important quotes translated into English. NVivo was used for data management and analysis 2 participatory videos | Indonesia | Understanding how leprosy patients deal with foreseeable difficulties | Meetings were generally held at participants' homes and one at the SARI's office. It was in collaboration with the community health centre. | Video Participants all had leprosy—8 persons with leprosy joined the first video, and 4 persons with leprosy and with impairments for the second. | Three different types of stigma emerged from the interviews: internalised, perceived, and enacted. Themes: 1. Impact on the participants—having a good time, a greater sense of togetherness, increased self-esteem, individual agency, and willingness to take action in the community 2. Challenges posed by impairments—Concealment was a persistent challenge, physical limitations, and group dynamics. |
| Peters RMH, Dadun, Lusli M, Miranda-Galarza B, Van Brakel WH, Zweekhorst MBM, et al. 2013 [16]. | One-to-one interviews- each participant was interviewed three times from exploratory to in-depth. 20 FGDs All interviews recorded, transcribed and translated into English—Data coded and analysed using thematic analysis using NVivo. | Indonesia | The views and experiences of people affected by leprosy and other key persons in the community. | At the SARI Office, the District Health Office, or a local hotel, Cirebon, Indonesia | 53 participants with leprosy— still under treatment or cured 20 FGDs—4–12 participants | Main themes: 1. Giving meaning to leprosy 2. Aetiology 3. Seeking care: Perspectives on diagnosis and treatment 4. Understanding healing and care 5. Impact of living with leprosy |
| Pelizzari V, de Arruda GO, Marcon SS, Fernandes CAM. 2016 [66]. | A qualitative study using semi-structured interviews. The interviews were recorded, transcribed, and analysed using content analysis in the thematic modality. | Brazil | The perceptions of people with leprosy about disease and treatment. | Interviews were carried out in participants' homes | 9 adults with leprosy, still undergoing treatment, mean age 57 years, 6 males. | Three categories: 1. Faced with the disease: the first symptoms to confirm the diagnosis 2. Motivations, benefits, and difficulties related to the treatment of leprosy 3. Family as support or exclusion |

(*Continued*)

**Table 4.** (Continued)

| Study | Methods for data collection and analysis | Country | Phenomena of interest | Setting/context/culture | Participant characteristics and sample size | Description of main results |
|---|---|---|---|---|---|---|
| Nasir A, Yusuf A, Listiawan MY, Harianto S, Nuruddin, Huda N. 2020 [62]. | Qualitative phenomenology using in-depth, face-to-face semi-structured interviews. All interviews were recorded, transcribed, and analysed using the interpretative phenomenology analysis approach. | Surabaya, Eastern Indonesia | The psychological experience of women living with leprosy in the community. | outpatients, Public health centre, Surabaya, Indonesia | 17 participants with multibacillary leprosy | 1. Mental stress is part of everyday life<br>2. Physical conditions that are far from aesthetic value<br>3. Guarantee of happiness is hard to come by<br>4. Looking for certain point to surrender to God for all suffering |
| Palmeira IP, Moura JN, Epifane SG, Ferreira AMR, Boulhosa MF. 2020 [65]. | A descriptive, qualitative research using semi-structured interviews and patients' medical records. Interviews were recorded, transcribed, and analysed according to Bardin's thematic content analysis. | Brazil | Leprosy patients' perceptions of their altered fundamental human needs and what they do to satisfy those needs | A health unit in Belem city, Brazil | 10 leprosy patients, 6 women, and 4 men, mostly MB. | The perception of altered needs and self-care<br>1. Physiologic needs—existence and survival of the individual related to oxygenation, mucosal cutaneous integrity, nutrition, hydration, etc<br>2. Security needs—physical and psychological security, involving health, safe work, social security, etc.<br>3. Love and/or social needs—life in society includes needs for meaningful living, respect, friendship, affection, etc.<br>4. Esteem needs—an emotional one which can cause changes to body/self-image<br>5. Self-actualising needs—spirituality, acceptance of facts, potential and problem-solving skills |
| Palmeira IP, Ferreira MD. 2012 [64] | An exploratory, qualitative study using semi-structured interviews. A field diary was also kept to record subjects' face and body expressions. The interviews were tape-recorded and transcribed and analysed using thematic analysis. Social representation theory was applied. | Brazil | The women's conceptions about their body that has been altered by leprosy | The outpatient unit, Specialised referral centre in health dermatology, Brazil | 43 women with bodily changes caused by leprosy | Themes<br>1. Aesthetic dimension of body The beautiful (healthy) and the ugly (sick):<br>a) Significant elements in the construction of representations<br>b) Functional dimension of the body<br>2. Strategies to overcome Living with one's new body and self-care |
| Nations MK, Lira GV, Catrib AM. 2009 [63]. | An in-depth ethnographic study using illness narratives, key-informant interviews, home visits, semi-structured interviews with physicians and patient observations of clinical consultations. Interviews were recorded and transcribed verbatim and analysed using thematic analysis. Contextualised semantic interpretation is used to link individual experience with systems of significance. | Brazil | The moral experience of leprosy patients with severe multibacillary leprosy | Two public health clinics in Sobral, Brazil, Home visits | 6 patients with MB leprosy, Patients' family members, neighbours, community healthcare workers, healers, physicians, other key informants- nurses, clinic staff, MoH staff etc | 36 topics emerged; eight themes:<br>1. denial of leprosy<br>2. social suffering<br>3. physical and social disabilities<br>4. everyday coping strategies<br>5. lay care practices<br>6. clinical approaches<br>7. patient-treatment approaches<br>8. health education interventions<br>4 stigmatising leprosy metaphors<br>1. repulsive rat's disease<br>2. racist skin rash<br>3. biblical curse<br>4. lethal leukaemia<br>Skin Spot Day: Educate or Discriminate |

*(Continued)*

**Table 4.** (Continued)

| Study | Methods for data collection and analysis | Country | Phenomena of interest | Setting/context/culture | Participant characteristics and sample size | Description of main results |
|---|---|---|---|---|---|---|
| Ebenso B, Newell J, Emmel N, Adeyemi G, Ola B. 2019 [91]. | Document reviews were done: historical materials from books and articles from major databases, leprosy control policies, records, and reports, etc. Semi-structured interviews with patients affected by leprosy and community members. All interviews were audio-recorded, transcribed verbatim, and coded for manual synthesis and categorisation. | Western Nigeria | How leprosy shaped the lives of leprosy patients in Western Nigeria, influenced by sociocultural context and organisational policies and practices. | A leprosy settlement in Kwara and Oyo States in West Nigeria, Interviews with patients were done in their homes. | People affected by leprosy n = 21, Women n = 11, men n = 10, community members n = 26 | Connotations and significance of leprosy: 1. the perception of leprosy as the most shameful and detested condition, 2. symbolic association with filth and immoral behaviour 3. Causation and transmission of leprosy: a) supernatural and b) natural causes: hereditary, air-droplet infection, contact with skin lesions and ulcers, and sharing cups, plates, and bedding with leprosy patients. 4. Document analysis—Four sources of stigma: a) cultural beliefs about leprosy b) health promotion messages in primary school books, c) religious teachings about leprosy d) campaigns by the leprosy service in 1950s Perspectives on symptoms and signs: missed early symptoms and signs lead to late detection as failed to associate to leprosy without deformities and inflamed reddish skin lesions 5. Cultural understanding of treatment and cure: The Yoruba people believe that leprosy can be treated but not cured as visible deformities cannot be restored to normal. 6. The changing stigmatisation of people with leprosy in West Nigeria—Five facilitators of acceptance: antileprosy treatment, good moral character, supportive family networks, livelihoods, and contribution to community survival. |
| Calcraft JH. 2006 [46]. | Qualitative study using semi-structured interviews, using gender-specific interviewers with translators. All interviews were recorded and transcribed and translated. The data was analysed using a simple indexing system with thematic analysis. | Nepal | The effects of the stigma of leprosy on income generation and loss. | Laldagh Leprosy Services Centre, meeting locations of the self-help groups at the hospital and the participants' homes. | Participants n = 19, men n = 9 and women n = 10, Leprosy affected person or a family member/close friend. | 1. Income was mainly affected by their physical disability. 2. Stigma accounted for decreased income in 7 of the cases. 3. The greater impact of income loss was due to the physical effects. 4. There was no evidence for courtesy stigma to cause decreased income. 5. Stigma was more evident when there was physical deformity. |
| van 't Noordende AT, Aycheh MW, Schippers A. 2020 [83]. | A cross-sectional design, qualitative approach, semi-structured interviews with individuals and FGDs. All the interviews were conducted in the local languages, audio-recorded, transcribed and translated into English, and coded. Open inductive coding and content/thematic analysis were carried out. | Ethiopia | Impact of leprosy, podoconiosis, and lymphatic filariasis on family quality of life | The interviews were conducted either in participants' homes or, if they were members of a patient organisation, in a private space near the patient organisation. They were based in 3 districts in the Awi region, Ethiopia | Participants n = 86, IDIs n = 56, and FGDs n = 30 made up of patients and their families. leprosy n = 12 leprosy and filariasis n = 1, and family members n = 17. | Themes: 1. Physical: symptoms, cause, and self-care 2. Psychological aspects and mental wellbeing 3. Level of independence: Day-to-day life, work, and resources 4. Environment: Attitudes and social participation 5. Social support and family relations |
| Van'T Noordende AT, Lisam S, Ruthindartri P, Sadiq A, Singh V, Arifin M, et al. 2021 [92]. | A mixed-methods study with a qualitative component using semi-structured interviews with individual participants and FGDs. The in-depth interviews and focus group discussions were audio-recorded, transcribed verbatim, and translated into English. The analysis was done using open, inductive coding with content and thematic analysis. | India and Indonesia | Leprosy perceptions and knowledge in India and Indonesia | Interviews were done at or near their homes. They were done in Uttar Pradesh, India, and East Java, Indonesia | A total of 110 participants (52 in India, 58 in Indonesia) were interviewed in-depth (45% female), and in India, 60 participants were included in seven focus group discussions | 1. Differences and commonalities in knowledge, 2. attitudes and practices regarding leprosy 3. Determinants of leprosy knowledge and leprosy-related stigma |

**Table 5. Summary of findings and categories for each synthesised finding.**

| Number of findings | Categories | Synthesised Finding |
|---|---|---|
| 13 findings | Bio-medical effects of leprosy | Biophysical Impact |
| 16 findings | Access to healthcare | |
| 14 findings | Functional impairments and disabilities | |
| 37 findings | Social effects of leprosy | Social Impact |
| 19 findings | Coping strategies | |
| 14 findings | Impact of leprosy on women | |
| 4 findings | Institutionalisation | |
| 19 findings | Meaning/knowledge of leprosy | Mental and Emotional Impact |
| 24 findings | Psycho-affective effects of leprosy | |
| 13 findings | Employment and economic effects of leprosy | Economic Impact |
| Total = 173 findings | Total = 10 categories | Total = 4 synthesised findings |

**3.3.1. Synthesised finding 1: Biophysical impact.** This synthesised finding consisted of three categories and 43 findings from 27 studies. [12,16,18,43–45,48–50,52,54,58–60,62–67,72,76,78,81–83,87] This synthesis documented the effects of leprosy on the skin, eyes, wounds on their limbs, nerve injury, and pain. Due to poor access to healthcare and delays in treatment, the individuals suffering from leprosy developed deformities that impacted their day-to-day activities. [12,16,44,50,52,59,72,81,87] Some studies stressed the importance of participating in self-care programmes. Due to the lack of information and education, the leprosy-affected individuals could not participate in the self-care programmes to improve their health outcomes. [44,50,60,64,65,76,81,87] All the studies showed that people with leprosy are faced with many challenges to get diagnosed and treated. Below is the explanation of the categories included in this synthesised finding.

*3.3.1.1. Category 1*: *Biomedical effects of leprosy.* There were 13 findings in this category that were extracted from 12 studies. [18,43,45,50,58,62–64,66,81–83] They discussed the physical symptoms and signs of leprosy and their difficulties. Some studies described the disfiguring spots, skin rashes, and ulcerations [62,63,66,82,83], deformities of the face and limbs [18,43,50,58,81], constant pain [64], or the lack of it. [45] One study reported the ignorance of the person who was unable to recognise the symptoms of leprosy [50]; in contrast, three studies showed a high suspicion of leprosy with the onset of the spots and rashes [66,82,83]. Some individuals reported feeling ugly because of the changes in their bodies. [58,62,64]

*'[. . .] a scabies-like rash on my whole body and it was itching; finally, the wound started from my foot and spread to my whole body, then a feeling of senseless, finally it eats my fingers, and I lose my fingers [...].' (Man affected by leprosy, age 60, page 6)* [83]

*Jung and Yang* [58] *reported an individual saying 'Back then, I had eyebrows, hands, and feet [. . .] People didn't recognize me because my face became swollen and my appearance changed'. (#2, page 4)*

*'I did not know it is serious. I got something like blisters from burns from fire. I did not give much attention to the wounds. My friend recommended me to come here. Now I have wounds as well as deformities.' (LalP1, page 5)* [50]

*3.3.1.2. Category 2*: *Access to healthcare.* Sixteen findings from 12 studies [12,16,44,50,52,59,63,66,72,76,81,87] supported this category. They highlighted the lack of health facilities near their homes [50,59], the cost of treatment [52], and not having their health

needs and concerns addressed in a timely manner [72]. The studies showed that the lack of knowledge regarding leprosy led to late presentations [50,59,87] and poor compliance with treatment [12,63]. Some reported the benefits of treatment and self-care, while others suffered some difficulties. [12,44,45,50,59,76,81] Many reported a good relationship with the healthcare professionals, who provided enormous support [16,59,66], but some found their rapid turn-over lacked continuity and caused insecurities [87].

> *'I used to live in (a small city) [. . .] I was there for five years but they didn't know what it was. I couldn't even walk and was in a lot of pain for five years. Only after a biopsy in [a bigger city] they found out.' (P1, Brown, page 7)* [72]

> *'It is difficult in the countryside. There is a lack of awareness. Health care professionals need to visit us at home because it is hard travelling to clinics.' (005-Patient, page 11)* [59]

> *'Anytime we were referred to Korle-Bu, we felt uncomfortable because we would be asked to buy our medication [. . .] with our condition, we cannot work and therefore cannot afford to buy medicines. We feel more comfortable with the doctors and nurses at the Leprosarium's health facility, where services are free.' (Female patient with visible deformities, page 66)* [52]

In another study, Da Silva and Paz [87] highlighted that the turnover of the health profes-sionals undermined the care given to these leprosy-affected people as some of them did not comply with the guidelines for supervision of the monthly MDT dose. *'It wasn't the same doc-tor, and that always complicated things a little. The nurse that I knew left, but I don't know where she went! So, the other people were a little lost because the right thing would be for the nurse to watch you take the medication. [. . .] Sometimes, I didn't even take it there; I took it at home.' (Man, 56 y, page 177)*

> In contrast, some nurses provided guidance about the disease and treatment. *'It means everything! She is very important, because she cares about her patients, cares about asking the agent to visit our homes, cares about asking the agent to call us to remind us to take the medi-cine [. . .], to complete the treatment.' (Woman, 24y, page 178)* [87]

*3.3.1.3. Category 3*: *Functional impairments and disabilities.* Functional impairments and activity limitations associated with leprosy were well described. Fourteen findings were extracted from 10 studies. [44,45,48–50,54,60,64,65,78] All the studies described the deformi-ties related to leprosy resulting from sensory loss, which was common, associated with muscle paralysis and trauma. They were the result of delayed treatment, causing complications. [44,45,50,54,60,78] Some of the leprosy-affected individuals also reported having issues with their sexual function. [48] Some of them believe that physical exercise is good but must be selective in their activities to minimise injuring the affected limbs. [49] Some studies reported how they adapt to lessen the effects of leprosy. [44,64,65]

Araújo de Souza and colleagues [44] reported a patient saying *'I used to do things that I can-not do anymore, I cannot work, I don't have any strength in my arms anymore nor in my legs like I used to have.' (Male patient with leprosy, page 512)*

> *'I'm not worried about my disease. I am worried about my foot. Without it, I could not work like other normal people.' (Female, age 60, unemployed, page 64)* [78]

Correia et al. [50] reported patients with manual jobs had difficulties due to complications of leprosy. *'I have difficulties holding items. My wounds make it difficult to plough land.' (lalP2, page 5)*

Carvalho et al. [48] studied the perception of sexuality of people living with leprosy. *'It gives me some cramps, and I feel weakness in the nerve. . .I think that this ordeal of sexuality comes from the nerve, and the disease is more in the nerve.' (D7, page 463)*

**3.3.2. Synthesised finding 2: Social impact.** The social impact consisted of four categories and 74 findings and was meta-aggregated from 38 studies. [12,14,16,18,34,43,45,47,52–58,61–67,69–78,80–84,87] This synthesised finding captured the interactions and relationships of the leprosy-affected people with others. The included studies showed they were isolated and excluded from society; hence their social problems were multifaceted. The social burden of leprosy included restrictions in social participation due to being different and causing feelings of guilt and shame. [4,18,73,83,93] The studies highlighted the interactions and challenges individuals with leprosy faced, including the problems faced by women in their relationships with their families and community. [54,57,62,64,69–71,84] Some mothers found it hard to be separated from their babies for fear of transmitting the infection. [47,71] The small number of studies that discussed gender did not focus on the problems faced by men. Fears of disease, stigma, and discrimination made life difficult for the affected people. Many were forced to hide their disease due to the unfounded beliefs of family, friends, neighbours, and the general public. [45,52,62,63,65,67] However, those who received support from family and friends had positive effects, although it gave them a sense of being a burden. [12,61,66,69,70,74]

Two studies showed that having money and the ability to contribute to the community made them more independent. It changed the way their families and community accepted them. [34,53] Institutionalisation was a way some countries adopted to isolate and stop leprosy-affected people from spreading the disease to others. [52,58,71,75] Following the JBI meta-aggregation process, the coping strategies of leprosy-affected people will be discussed in this category. However, it could also be linked to mental, emotional, and biophysical impacts. The way people coped with leprosy depended on their understanding of the disease, its complications, and its treatment. It affects how they seek help from healthcare services, gains support and understanding from their families and community, and their adherence to treatment and self-care. Non-Governmental Organisations also play a part in helping people cope with leprosy. [82]

*3.3.2.1. Category 4*: *Social effects of leprosy*. This was the largest category with 37 findings extracted from 26 studies [12,18,34,43,45,47,52,53,55,61–63,65–70,72–75,77,78,80,83] that highlighted the social consequences of leprosy: the stigma and discrimination. The studies showed that leprosy-affected people have a high social burden and suffer stigma and exclusion. However, some studies have shown they could change their status in society by getting appropriate treatment, taking their medication, and participating in economic reforms. [34,53] Unlike some illnesses where the patients struggled to convince others that they were ill, leprosy-affected people had to convince their families and community that they were no longer infectious. [52,75] The studies showed that some people often felt unsupported by their family and friends and accepted the social isolation frequently featured in their experience. [18,43,62,69,73,75,76,78,83] However, some reported the kindness and concerns of their families and community. [12,66,69,74]

Dako-Gyeke and colleagues [52] reported that many people affected by leprosy experience stigma and discrimination through their interactions with people. *'Many people, including my family members, do not believe I am cured because of the deformities, especially the sores on my leg. . .' (Male with visible impairments, page 64)*

*'If there are no urgent matters, we prefer staying at home. From our experience, it is better to not make contact with people. People surrounding us expressly avoid us, and they continually label us as patients with a contagious disease.' (Person affected by leprosy, page 6)* [18]

Socio-economic rehabilitation (SER) has helped to reduce stigma and stimulate change in people's attitudes, leading to the inclusion of people affected by leprosy. The individuals feel valued and accepted by society and can contribute to their community. *'It has improved my access to places. Where I couldn't go before, I can now go; you know, 'money is the vehicle of interaction.' I am now going to places like church and contribute my own share.' (P3, male participant, page 104)* [53]

Contrary to what some studies reported about stigma and discrimination, Ebenso et al. [34] highlighted that in some cultures, leprosy-affected individuals were welcomed back by their families after their discharge from treatment. Their communities were willing to accept those who had completed their MDT despite their deformities. *'[. . .] I was shunned before I received treatment for leprosy. No one invited me to participate in family or communal activities because they thought I would infect them with leprosy. But after I was treated and discharged from MDT, they have welcomed me back into the family. I was also recalled by the community. Now we all participate together whenever there are ceremonies such as marriages.' (Female, Food vendor affected by leprosy, page 8)*

Pelizzari et al. [66] showed how important it was to people with leprosy when their families showed interest and gave them care and attention while receiving treatment. *'My daughter thought I was not sick because she said that I ate well. Then she saw I was crying in pain and said it was serious and started running with me in the consultations. At first, I lived alone; then, my daughter came to live with me. She was watching me at first because I could not walk with pain.' (E4, M, 77 y page 470)*

On the other hand, some tried to exclude them even if they were no longer contagious. *'A brother treated me differently, and I felt very offended. He said it was to separate plate, cup, forks, spoons [. . .] I said I was not contagious.' (E7, F, 44y, page 471)* [66]

*3.3.2.2. Category 5*: *Coping strategies.* There were 19 findings extracted from 16 studies [12,14,16,47,54–56,61,62,65,67,76,78,81,82,87] demonstrating the different strategies used to cope with the leprosy burden including concealment of their disease. [14,54,87] The studies showed that patient education is integral to helping them cope and manage their lives. [12,55,67,70,76,81,82] One important coping strategy is social coping, when those suffering from leprosy can request support from their families, friends, and community. [16,56,62,69,71,78] Another helpful strategy is to find ways to maintain their emotional well-being by changing the way they think about the problems and accepting them [18,78,81,82], turning to their religion and spiritual beliefs to cope with the disease. Other coping strategies would include meaning-focused by gathering information and understanding them or problem-focused when they can ask for help and support from the experts and get treated. [12,56,67,76,78,81,82] These coping mechanisms assist people with leprosy adjust to the challenges of their disease while helping them manage their lives and maintain their emotional wellbeing. [29,36]

Heijnders [55] highlighted how the people affected by leprosy would try to conceal their disease for as long as possible to keep their place in the community and avoid negative social consequences. Felt stigma led to a strategy of concealment. *'My disease is not clear and that is why most of them could not find out.' (Patient with leprosy, page 441)* The diagnosis of leprosy was seen as a trigger to discrimination. *'I did not tell my family about the disease. I was afraid they would be tense and get worried. We do not have enough food to eat and clothes to wear. I thought that whatever happens, will happen to me only.' (Man with leprosy, page 442)*

*'You can't be completely open with some people that you have this, because they think you'll transmit the disease to them.' (Woman, 24y, page 179)* [87]

The most dominant aspect of social impact is the acceptance and support of family and the community with the leprosy-affected person's changing health and disabilities. Jatimi and colleagues [56] showed that this allowed the person to accept his disease and the challenges and take appropriate actions to continue his daily activities. *'Thank God, the family supported me during the treatment, taking me to get medicine. The neighbors also did not change their attitude; the officer always visited me. Sometimes delivering drugs.' (R1, page 99)*

Peters et al. [16] showed that many people affected by leprosy believe that God could be involved in them contracting leprosy. They view leprosy as a challenge from God to make their faith stronger. It helps give meaning or an understanding of why they get leprosy and find strategies to cope with the disease. Similarly, Nasir et al. [62] also reflected on the religious approach of the person with leprosy for the test they believed was given by God. *'God gave the disease and I believe God also gave the medicine. For me, this is a test that I must go through, and I still have to try so that I can get the medicine that can cure my disease.' (P. 13, page 309)*

Health education activities are helpful to control and minimise the damage of leprosy. Through community education, the nurse or healthcare workers inform people with leprosy how to recognise the early symptoms of leprosy and get treatment without delay to prevent complications to their skin, face, and limbs. This will empower them and give them the confidence to look after themselves. [81] *'The more people have information, the better. It facilitates treatment adherence. You know, the more information, the better.' (José, page 2819)*

Van Netten et al. [78] found that the local community believed leprosy is an infectious disease that can be cured with medicines. They seemed confident in the medicine provided by the local health workers to treat leprosy. *'I suggest to people it's just a disease, you must go for treatment on time; otherwise it will infect other people in your family too. It will make you disabled. If your family gets the same symptoms send them to Lalgadh as soon as possible.' (Male, age 60, page 65)*

*3.3.2.3. Category 6*: *Impact of Leprosy on Women.* A gender approach can help health professionals understand the challenges women with leprosy face. They have greater concerns about their physical appearance [64], negative emotional response [54,58,64,84], economic independence [53], and limited social participation [84]. However, self-help groups have helped boost their confidence in themselves. [57] Fourteen findings were extracted from 6 studies [53,54,57,58,64,84], highlighting the impact of leprosy on women.

Palmeira and Ferreira [64] aimed to understand how women conceive their bodies that have been changed by leprosy, particularly when they identify themselves in terms of their physical appearance. *'[. . .] how I see my body now? Totally unrecognizable. I cry when I see my body like this, because I didn't have these ugly things before, and now I do.' (I22M, page 382)*

*'It's difficult for us to talk because it's not everyone who accepts it, there's a prejudice right [. . .] I did not want my family to suffer [. . .] Then only my boss who knows, my manager, that I told for them and they said that they would help me, that I did not need to tell my co-workers, that it could be between us.' (Interv 6, page 663)* [54]

Ebenso et al. [53] showed how socioeconomic rehabilitation could improve the self-esteem of leprosy-affected people by allowing them to be engaged in productive work and fulfilling their social roles. *'It has increased my dignity because I am now able to send my children to school, and I am able to work like other women in society.' (P1 Female, page 105)*

*3.3.2.4. Category 7*: *Institutionalisation.* In the early days of leprosy treatment, people with leprosy were removed from their homes to the leprosaria in order to stop the transmission of the disease to the rest of the community. There were four findings from 4 studies [52,58,71,75]

that highlighted their experiences of living in isolation separated from their families. They lost contact with their families and friends and the outside world. [52,58,71,75]

Dako-Gyeke et al. [52] found that many participants had suffered stigma and discrimination from their families and community and had no intentions of leaving the Leprosarium, even if they were cured. They felt safe and comfortable being among people who were like them. *'I have no family outside this Leprosarium. Because of our visible impairments, many people do not want us around them, especially living with them in the same house [. . .] even if we had money to pay rent, landlords were not willing to accept us since our own family members were unwilling to accept us. We will continue to reside in this Leprosarium because we are happy here.' (Female with visible impairments, page 69)*

The compulsory hospitalisation for the treatment of leprosy was not just for the treatment offered but to get away from the cruelty and persecution they faced from their community. [58] *'I was living hidden in a small room, but the local police came and said that there is a good place where they could give me medicine and treat my disease, so I left the house with them to be hospitalized.' (#9, page 5)*

Many people who have been cured of leprosy preferred to continue living at the leprosarium as they identified with each other as family, which fulfilled their need to belong. [75] *'This place has become a home for me because I can't live comfortably in my hometown and I feel better living among my likes here. Here, we eat and do many other things together without anyone treating the other differently.' (Patient cured of leprosy, page 160)*

**3.3.3. Synthesised finding 3: Mental and emotional impact.**   Leprosy affects the individual's mental, emotional, and spiritual wellbeing. This synthesised finding covered their understanding of leprosy, including the myths and beliefs regarding causation and transmission of the disease. [12,14,16,19,34,44,50,59,63,65,68,70,72,78,79,82] The studies also described the impact of leprosy on their feelings and their interactions with the people around them. [19,45,47,49,50,52,56,61–63,65,67–69,73,75,77–80] Due to poor public education, people all over the world still have incorrect and harmful beliefs about leprosy. [12,14,16,50,59,70,72,79] They caused the people inflicted with leprosy to hide their symptoms leading to delays in treatment.

Studies have shown that leprosy-affected people felt guilty and ashamed of themselves [47,56,61,67,75,80], and the studies reported them suffering from depressive and anxiety disorders. [47,52,69,73,79] The emotional impact of leprosy is commonly related to low self-esteem [65], feeling fearful [56,69], stress [50,62], and frustration [49]. Learning to accept the disease and recognising their limitations allowed them to cope better and adapt their lives to the disabilities. [94]

This synthesised finding was made up of 2 categories and 43 findings. People need to know that leprosy is curable and that the treatment is free. [95] The long duration of the illness, the physical disabilities, and the rejection by their families and community increase their risks of psychiatric disorders. [21]

*3.3.3.1 Category 8: Meaning/knowledge of leprosy.* A significant number of people get infected with leprosy every year, but their beliefs and misconceptions about leprosy prevent them from getting treatment in a timely manner. There were 19 findings extracted from 16 studies [12,14,16,19,34,44,50,59,63,65,68,70,72,78,79,82] in this category.

Correia and colleagues [50] showed that many people have alternative beliefs about leprosy, from curses and spells to what they or their forefathers did or did not do. *'Leprosy I am not sure. Maybe it is a curse; it is due to any bad deeds in my previous life.' (LalP4, page 6)* Some attributed it to their work, the food, or the temperature. There is a general lack of knowledge regarding leprosy contributing to delayed treatment. *'As I was working in Madras, where the*

*temperature is very high. I also ate a lot of spicy food. I believe my disease is due to this, the heat and spicy food.' (LalP10, page 7)*

There was a general lack of understanding and awareness about leprosy among the people with leprosy and their carers. They were unable to identify the cause of leprosy. *'I never shower with hot water and then suddenly with cold, or after eating, because I thought that's what causes leprosy. I don't know how I got it.' (018 –Patient, page 17)* [59]

Some participants felt that public education would help to reduce misunderstandings about leprosy. [72] *"I think they should do more than just talk. They speak about leprosy on the radio, and nobody cares because they don't know how to understand it [lack of education of population]. The lack of education is the problem." (P99, White, page 7)*

*3.3.3.2 Category 9: Psycho-affective impact of leprosy.* There were 24 findings from 20 studies [19,45,47,49,50,52,56,61–63,65,67–69,73,75,77–80] highlighting the psycho-affective impact of leprosy. It has been shown that the negative feelings and attitudes of people affected by leprosy can affect their mental wellbeing.

Ramasamy et al. [69] highlighted the negative behaviour towards people affected by leprosy in the community, causing them to hide their disease due to fear of adverse community reactions. *'I was very upset and depressed and had suicidal thoughts many times due to this disease. Nowadays I get unnecessarily stressed and afraid to tell anyone. ' (Female, 33 years, page 70) 'Due to the visibility of patches on my face and hand, all of them came to know I was affected with leprosy. But myself getting angry and frustrated to live, asked myself, Why me? What sin have I committed?' (Female, 40 years, page 70)*

Similarly, Jatimi and colleagues [56] reported that psychosocial problems in people with leprosy are common, characterized by their feelings of fear about their changing bodies. *'I feel ashamed, not confident to gather with neighbors after leprosy because my body's skin has turned black.' (R10, page 98)* They also suffer anxiety, sleeping difficulties, and withdrawal from the community due to their physical disabilities. *'I often have difficulty falling asleep at night, sometimes fearing that the disease will get worse. When I was first told that I had leprosy, I always felt afraid I would not recover from this disease.' (R6, page 98)*

Dako-Gyeke et al. [52] illustrated that suicide was often contemplated by those with leprosy due to the stigma, discrimination, and neglect they faced. *'My family did not care about me, even getting a place to live and food to eat was a problem [. . .] I became very sad, bad thoughts came to my mind and I felt like ending my life [. . .].' (Male participant with visible impairments, page 65)*

**3.3.4. Synthesised finding 4: Economic impact.** People affected by leprosy suffer from pain, weakness, and activity limitations even if they have completed their treatment. Many suffer from stigma and discrimination and have difficulty securing work because of their visible deformities. [18,43,46,59,70] As a result, they try not to reveal their illness to their employers and work colleagues for fear they would lose their jobs. Even those employed might need a change in their job specification, impacting their employability. Some employers are able to compromise and make some adaptations or provide skills development. [54,56,83] Three studies showed that people with leprosy could become financially independent by giving them micro-loans with the introduction of the socio-economic rehabilitation. [51,53,57]

This synthesised finding was made up of 1 category and 13 findings. The studies showed that their unemployability has a tremendous economic impact on their families and community. [51,52,70,72] Many of them live in poverty as they experience loss of income and high cost of treatment. Some of them are lucky to get some form of financial assistance, but many beg on the streets as either they do not receive any financial support or it is not enough for them to live. [52]

*3.3.4.1 Category 10*: *Employment and economic impact of leprosy*. There were 13 findings extracted from 13 studies. [18,43,46,51–54,56,57,59,70,72,83] The studies showed how the people with leprosy reflected on how the disease affected their ability to work.

Dako-Gyeke et al. [52] highlighted that the difficulties faced by leprosy-affected people to find jobs were due to their physical disfigurements and deformities and the negative perceptions and beliefs people had about the disease. So, to support themselves financially, many beg on the streets to get some money. *'The money we receive from government is not sufficient. This is a big problem because I do not have anybody to support me financially [. . .] If I wake up in the morning and I have no money on me, I borrow one Cedi to pay for my lorry fare to Lapaz or Nya-mekye (communities near the Leprosarium) to beg for alms for two or three hours, and I get some money [. . .].' (Male, with visible impairments, page 67)*

Jatimi et al. [56] showed how people with leprosy could accept and make positive adaptations to their physical conditions by taking up gardening or working in the fields. They can help improve their mental and physical health. *'Now, I work in the paddy fields of my father. The results are quite good. If the harvest is good, it can benefit a lot. [. . .]' (R3, page 100)*

Self-help groups have been a great success story in the communities where they are active, mainly in reducing the stigma of leprosy. They help empower women to live independently and have equal access to opportunities and resources. [57] *'After taking 24 months of medica-tion, I became cured. I have started income generation activities effectively. Initially, we were given Nrs. 20,000/-. Now we have about 500,000/-profit. People are happy with me to see my progress and that I repaid loans as well[. . ...]." (Woman, 55y, page 163)*

Goncalves et al. [54] demonstrated how work and daily activities had to be adapted to changes caused by leprosy. *'So, when they knew, they immediately sent me to the work doctor. And the doctor at work immediately removed me from handling food [. . .] Then they put me in the refrigerator sector, which is where the yogurts are, those things there [. . .] That is food that is already packaged, you know?' (Inter 3, page 663)*

However, as shown by Lusli et al. [18], not everyone is lucky, as leprosy-affected people still face challenges finding a job. *'When I attended an interview for recruitment, the interviewer noticed the skin patch on my face. I told them I am cured from leprosy, but they did not trust me. I even showed them a formal letter from the community health services, but they rejected me.' (Person affected by leprosy IDI 1, page 4)*

## Discussion

We systematically reviewed and meta-aggregated the experiences of approximately 1209 leprosy-affected people to obtain a rich understanding of their lives and how they managed the disease. The qualitative and mixed methods studies were from twelve countries, mainly Brazil, Indonesia, and Nepal. We extracted 173 credible findings with illustrations from the 49 included articles. [12,14,16,18,19,34,43–84,87] They were synthesised into ten categories and subsequently meta-aggregated into four synthesised findings to highlight the impact of leprosy on the lives of those affected: biophysical impact, social impact, mental and emotional impact, and economic impact. These findings are not new, but they are consistent across the included studies from a patient's perspective. The review has allowed us to understand their vulnerability and the debilitating negative consequences of the disease.

This review has clearly demonstrated that leprosy has a multi-domain effect on the affected individuals. The World Health Organisation defines *health* as more than just physical health but involves other domains like emotional, social, intellectual, environmental, financial, and spiritual health. [96] The residual, permanent, and unresolving physical disfigurements, disabilities, and painful neuropathy of leprosy continue to perpetuate the psychological, social,

and economic impact of the disease. The neurological impact of leprosy is the main symptom of the disease that affects individuals long after eradicating the bacteria. Neuropathic pain is now known as a long-term and late complication of leprosy. [97]

Like most chronic illnesses, this review has shown the tremendous impact of leprosy on the individuals' perceptions of the disease and its effects on their thoughts, self-identity, confidence, and everyday functioning. [98] The analysis of the studies in this review showed that most individuals continued to face challenges at different points of the disease process as they endured the disabling complications of leprosy and were unable to return to their past level of involvement in society. [18,46,59,70,81] They showed that those who accepted and recognised their limitations coped better with their disabilities. [27,44,64,65,99] Hence, health professionals must consider the affected individuals' illness perceptions to provide them with the right support. Leprosy would have changed their lives, so studies in this review recommended appropriate education to develop positive attitudes and empower them to look after themselves. [12,44,45,50,59,76,81]

Health education has been shown to reduce stigma, minimise disease burden, and help encourage leprosy-affected individuals to seek help and participate in self-care. [12,50,67,70,76,81,82] *'The more people have information, the better. It facilitates treatment adherence. . .'.* [81] Further research should explore ways to reach out to the disadvantaged people with leprosy in rural areas to provide vital health messages. Lepra, a charity organisation, and many other mobile health units have been working in remote communities to deliver health education on leprosy, through mobile phone technologies or through their activities, like screening films and community talks using posters, flip charts, and health pamphlets. [100] Digital technology and mobile phones can transform health and social care by providing access to health experts and effective support to the rural health workers and people in the less accessible areas to get timely treatment. [101] Radio shows in isolated communities may also be helpful. [19,72] Health information and support could be provided through school education programmes to screen the children. The healthcare workers could train the teachers to pick up the early signs and symptoms of the disease. [72,102]

Across the studies, a positive relationship between healthcare professionals and leprosy-affected people has demonstrated confidence for these individuals to participate in self-management, leading to treatment satisfaction and less treatment burden. [12,50,60,64,65,76,81,87] This strategy could be adopted for the future management of leprosy-affected people by arranging for them to see the same doctor or nurse for their repeated visits to foster better relationships and trust. Likewise, studies on chronic illnesses showed that the arrangement of seeing the same doctors for their patients resulted in good outcomes and improved the quality of care. [103,104]

Darkening of the skin affects the patients' appearance and may influence treatment adherence. It is a known side-effect of Clofazimine, one of the primary drugs used to treat leprosy, besides rifampicin and dapsone. [105] Although Clofazimine is very effective, the review showed that some people with leprosy might prefer not to take the medication for fear that others could identify they suffer from leprosy. [12,56,59,80] They should be informed that the pigmentation is reversible and will gradually fade when Clofazimine is stopped but may take several months to clear up.

While religious bodies have been active in providing care and spiritual support, religion can also reinforce some discriminations related to the misconceptions surrounding the disease, where leprosy has been degraded to a curse. [50,63,70,106] Besides the people's disappointment with God for giving them leprosy, this review also identified the role of religion to provide a level of comfort for the people affected by leprosy and help them adjust to the disease and its complications. [16,56,62,69,71,78]

Neglected tropical diseases, including leprosy, have common underlying causes based on poor living conditions and poverty. Early diagnosis and treatment are used in leprosy and are the same for other NTDs like leishmaniasis, Buruli ulcer, and Chagas Disease. The World Health Organization supports an integrated approach to control skin NTDs. Hence, it is possible to integrate the training, active case detection, and management of the diseases, which will be timely and cost-effective. [107–109] Similarly, though different in aetiology, people with poorly controlled diabetes are also at risk of injuring their extremities and developing insensitive ulcers due to neuropathic foot lesions. Both these diseases can have nerve damage even at the time of diagnosis. [110] It is important to encourage these individuals to be involved in self-care. The prevention and management of neuropathic extremities in these two conditions should be guided along the same lines. It will help reduce stigma and might even be cost-effective to run combined clinics to treat them. The integrated clinics will be a step towards reducing leprosy-associated stigma. [93]

The review has elucidated the tremendous emotional burden of leprosy experienced by the leprosy-affected individuals and their families. This is attributed to the trauma of being diagnosed with leprosy, the long-standing nature of the disease, the associated physical disabilities, and the stigma associated with it. [21] Depression, anxiety, and suicidal ideations are commonly reported, and they suffer from low self-esteem and self-worth and loss of self-respect. [21] According to Govindasamy et al. [20], the mental health of people affected by leprosy has not been studied much. About 30% suffer from mental health issues, especially women and those who are poor, with low or no education, suffering from disability. We need to pay more attention to these issues to plan interventions to support them. Some of the studies highlighted the role of social networks, especially family support. The review showed that they were often rejected by their families and local communities. [52,56,61,62,73,75] The trained health workers can support these people experiencing psychological difficulties related to the disease. [16,50,62,69,78,80]

The economic impact of leprosy is significant due to the high cost of having the disease, as even though the treatment is provided for free, they still need to pay for transportation to the health centres. [23,25,53] Their physical disabilities and other associated symptoms cause their inability to work [43,46,59,61,70], leading to income loss and unemployment. [51,52,70,72] Consequently, many have to beg on the streets and live in poverty. [52,68,111] The review showed the discrimination they faced because of the visible impairments, which cannot be reversed even though they have been cured. [4,16,18,74] The children are also affected by the disease due to the loss of educational and future employment opportunities. (36,52–54,57) However, there has been some evidence to suggest increased success in behaviour change where interventions have focused on improving self-esteem, skills acquisition, and financial independence. Ebenso et al. [53] highlighted the impact of socioeconomic rehabilitation on stigma reduction. Similarly, Dadun and colleagues [51] showed that socioeconomic intervention could support leprosy-affected people stigmatised by the disease. This review suggests that greater attention should be placed on supporting people with leprosy to improve their economic status.

This review has shown the interconnectedness of the various effects of leprosy on the lives of those affected. The following chart (Fig 2) shows the relationships, and we believe this summary of the effects and impact can help guide other leprosy or chronic disease researchers to design their data collection instruments.

## Strengths and limitations

The meta-aggregation of the 173 findings into the four synthesised findings led to an overall understanding of the experiences of individuals living with leprosy compared with the fragmented results of individual studies. Although the four synthesised findings are not new, the

results will interest the researchers and all stakeholders studying leprosy, especially those involved in managing and supporting the individuals with the disease.

The literature search was conducted with the help of a Medical Information Specialist at Vrije University. A thorough search for eligible studies was carried out. It was decided to limit the studies to those that have been peer-reviewed because of the qualitative nature of the studies. Non-English articles without full translation into English were also not included because of limited resources available for comprehensive translational services. There was sufficient published literature bearing the individuals' experiences of living with leprosy. The reviewers hoped to maximise the methodological quality and reporting of the included articles. However, all studies not published in peer reviewed journals and from the grey literature were not included. It would probably have little impact on the overall results due to the quality of the included studies. This change was the only deviation from the original protocol.

We ensured the rigour of this review by using two independent reviewers to double screen the articles retrieved from the search to the inclusion of the full-text articles. We did the same for the critical appraisal, data extraction, data synthesis, and assessing the confidence of the findings. The reviewers discussed any discrepancies in their decisions to reach a consensus.

Some of the included studies lacked the researchers' positionality and reflexivity and possible influence on their studies. Multiple illustrations were used in this review to represent and support the data accurately. Although there were some gender-based studies, the focus was on the impact of leprosy on women, but there were none on men. This could potentially be investigated in future research.

There were limited studies that used a comprehensive participatory research approach, which would have given a deeper understanding of the lived experience of leprosy-affected individuals. Perhaps future research could adopt this method to understand their needs and concerns to improve their support.

## Conclusions

In this review, the findings were collated from multiple qualitative sources to synthesise the evidence for the experiences of individuals living with leprosy. It has conceptualised the four domains: biophysical impact, social impact, mental and emotional impact, and economic impact, and highlighted the diverse challenges. They confirmed the importance of studying the impact of disease using the biopsychosocial and economic model shown in Fig 3. It looks at leprosy from the biophysical and psychological aspects, merging into resilience and thriving and non-illness sectors, incorporating the social and economic domains. The methodological approach used in this review is translatable and can be reproduced to evaluate the impact on the health of other chronic and non-communicable diseases.

Crucially, in view of the multi-domain impact of leprosy on those affected, their families, and the community, a transdisciplinary research approach involving all stakeholders will be a holistic and participatory way to understand their problems. This technique will allow the participation of stakeholders to co-create the solutions or interventions to empower them to get diagnosed, treated, and self-manage their disease.

## Recommendations for practice or policy

The overall findings of this review were credible evidence rated as low or moderate, as shown in the ConQual Summary of Findings, following the JBI process (S7 Appendix). It is recommended to use the ConQual framework in the research design to improve the quality of the research using qualitative studies. Qualitative evidence synthesis will help us fully understand the impact of chronic diseases like leprosy on individuals suffering from the disease. The

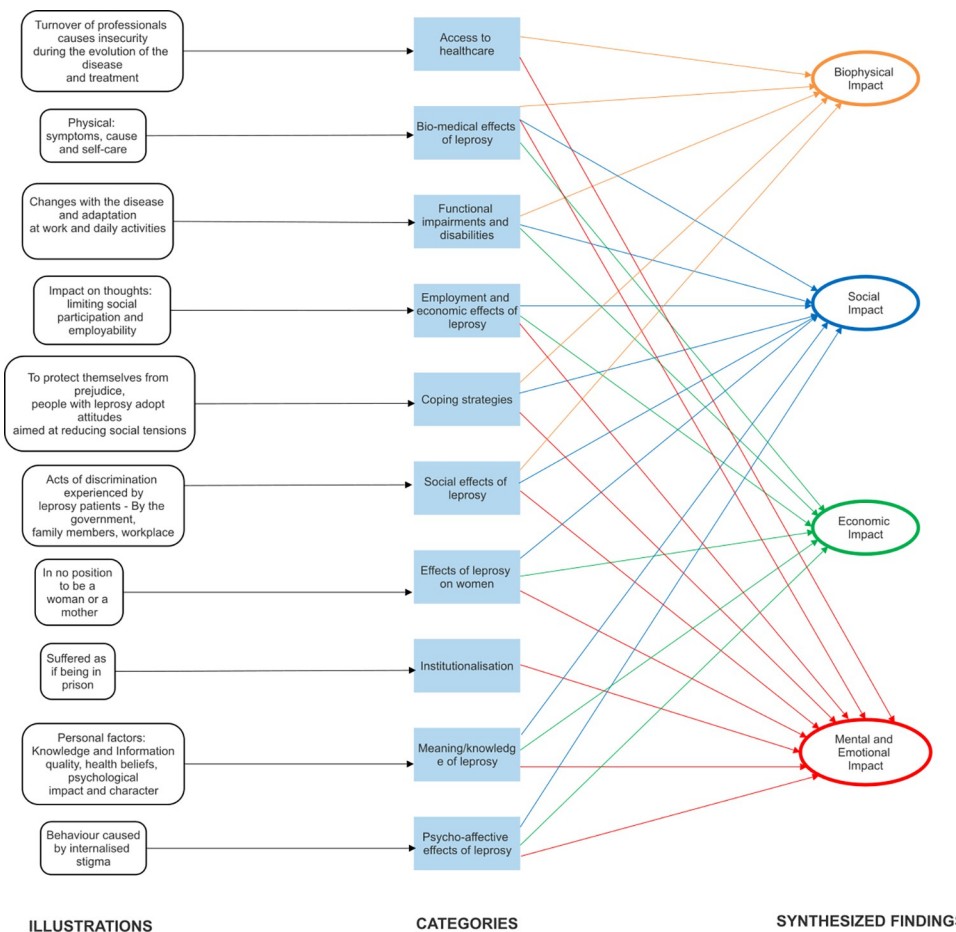

**Fig 2. The effects of leprosy on the lives of the leprosy-affected people.**

synthesised findings will provide evidence-based insights for the generation of policy and interventions targeted to overcome the impact of the disease on individuals and their communities and provide essential support for their wellbeing.

## Recommendations for research

Future researchers should ensure the inclusion of keywords in their titles and abstracts of their manuscripts to ensure visibility by search engines. Non-English articles should be considered in future reviews and may shed further light on the subject. Future research could also assess whether some of the strategies we discussed to encourage self-help will lead to improved health outcomes. We should also look at conducting primary research to study processes that could overcome the identified synthesised findings, such as strategies to provide skills development to improve the employability of leprosy-affected individuals.

The categories and synthesised findings of this review can be used as a framework for further qualitative studies on the lived experience of individuals with leprosy and other chronic diseases.

## Reviewers' reflexivity statement

Throughout the review process, the review team referred to the registered protocol (PROS-PERO: CRD42021243223), and they have explained the deviation undertaken. The authors

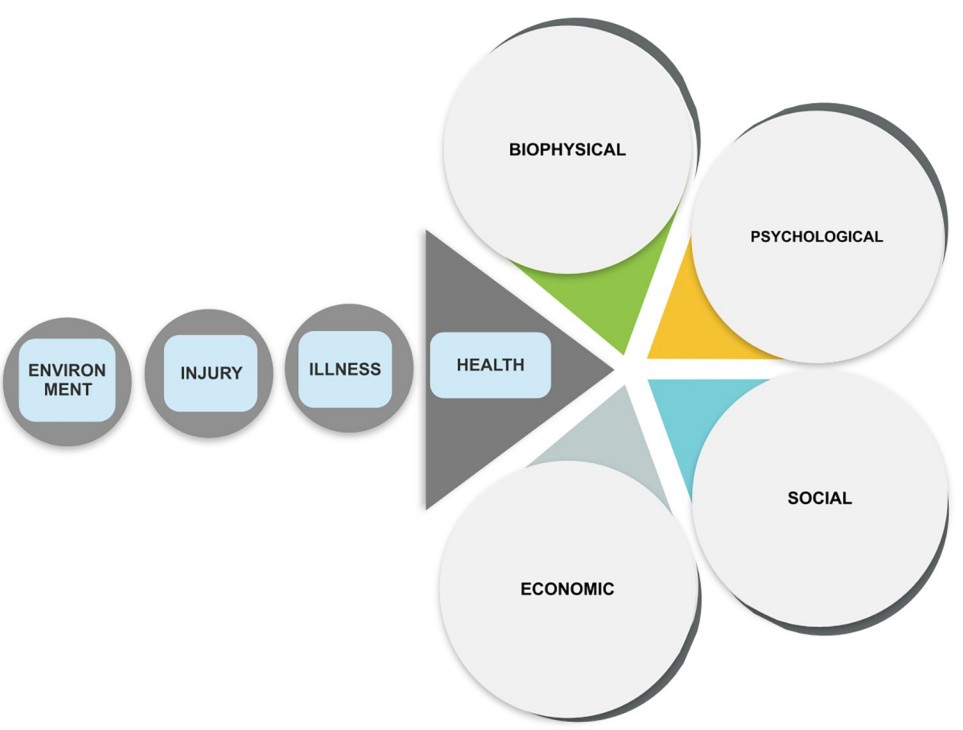

**Fig 3. Impact on health model—based on the four domains synthesised in this review.**

have experience in the subject matter and continually considered their views and possible influence on the review.

## Supporting information

**S1 Appendix. Overview of the search terms per database.**
(DOCX)

**S2 Appendix. Excluded studies.**
(DOCX)

**S3 Appendix. Included studies.**
(DOCX)

**S4 Appendix. JBI checklist in JBI SUMARI.**
(DOCX)

**S5 Appendix. Critical Appraisal Results.**
(DOCX)

**S6 Appendix. List of Study Findings with Illustrations.**
(DOCX)

**S7 Appendix. Summary of findings.**
(DOCX)

## Author Contributions

**Conceptualization:** Norana Abdul Rahman, Vaikunthan Rajaratnam, Ruth M. H. Peters.

**Data curation:** Norana Abdul Rahman, Vaikunthan Rajaratnam, George L. Burchell.

**Formal analysis:** Norana Abdul Rahman, Vaikunthan Rajaratnam.

**Methodology:** Norana Abdul Rahman, Vaikunthan Rajaratnam.

**Project administration:** Norana Abdul Rahman.

**Software:** George L. Burchell.

**Supervision:** Ruth M. H. Peters, Marjolein B. M. Zweekhorst.

**Validation:** Norana Abdul Rahman, Vaikunthan Rajaratnam, George L. Burchell.

**Visualization:** Norana Abdul Rahman, Vaikunthan Rajaratnam.

**Writing – original draft:** Norana Abdul Rahman.

**Writing – review & editing:** Norana Abdul Rahman, Vaikunthan Rajaratnam, George L. Burchell, Ruth M. H. Peters, Marjolein B. M. Zweekhorst.

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
