## [Decision Letter · Decision Letter 0]

8 Jul 2022

Dear Dr Abdul Rahman,

Thank you very much for submitting your manuscript "Experiences of living with leprosy: A systematic review and qualitative evidence synthesis" for consideration at PLOS Neglected Tropical Diseases. As with all papers reviewed by the journal, your manuscript was reviewed by members of the editorial board and by several independent reviewers. In light of the reviews (below this email), we would like to invite the resubmission of a significantly-revised version that takes into account the reviewers' comments. 

We cannot make any decision about publication until we have seen the revised manuscript and your response to the reviewers' comments. Your revised manuscript is also likely to be sent to reviewers for further evaluation.

Sincerely,

Alberto Novaes Ramos Jr

Associate Editor

Guilherme Werneck

Deputy Editor

Reviewer's Responses to Questions

**Key Review Criteria Required for Acceptance?**

**Methods**

-Are the objectives of the study clearly articulated with a clear testable hypothesis stated?

-Is the study design appropriate to address the stated objectives?

-Is the population clearly described and appropriate for the hypothesis being tested?

-Is the sample size sufficient to ensure adequate power to address the hypothesis being tested?

-Were correct statistical analysis used to support conclusions?

-Are there concerns about ethical or regulatory requirements being met?

Reviewer #1: Methods were clearly articulated. Details described in a way that others could use the same methodology in future studies and reviews. It is unfortunate that articles had to be excluded due to language.

Reviewer #2: The study design appropriate to address the stated objectives. The study employed well known methods of systematic reviews, adopting multiple electronic data bases. The strategies of data mining and selection were clearly and coherently described and were appropriately developed.

Reviewer #3: The study's objective is well-defined and consistent with purpouse.

In terms of study design, a meta-aggregated systematic review is deemed sufficient to achieve the stated goal.

The synthesis of qualitative evidence was carried out by the authors using the method advised by the Joanna Briggs Institute, and they were quite transparent about all the tools utilized, from data extraction to the evaluation process that was carried out at various stages of the study. This proves that the procedure was thorough and gives the text in question credibility.

**Results**

-Does the analysis presented match the analysis plan?

-Are the results clearly and completely presented?

-Are the figures (Tables, Images) of sufficient quality for clarity?

Reviewer #1: The results were clearly and systematically presented. Tables and Figure 1 were very helpful in summarizing results for quick visualization and review.

Reviewer #2: The analysis successfully presented match the analysis plan. The results clearly and thoroughly presented. The figures and tables were very helpful and clearly presented.

Reviewer #3: The results are presented in a clear and organized way. The data extraction table presented provides reliability to the data extracted from each study, as well as the statements presented throughout the text that are supported by the table in the appendix.

In addition, the way in which the results are analyzed meets the purpose of the study, which is to aggregate the results found in the primary studies.

The results obtained are easily identified according to the organization of the text and are presented in a didactic way through the proposed table and figures.

**Conclusions**

-Are the conclusions supported by the data presented?

-Are the limitations of analysis clearly described?

-Do the authors discuss how these data can be helpful to advance our understanding of the topic under study?

-Is public health relevance addressed?

Reviewer #1: It was surprising to see that few studies were included from India in the analysis. As India is one of the countries with the greatest number of leprosy affected persons, the authors may want to comment why so few studies from India were in analysis and make a recommendations.

Check Figure 2: The effects of leprosy on the lives of leprosy-affected people helped the reader visualize the Impact of each category along with an illustration. After reading I was surprised that the Biomedical effects(2/10) of leprosy and functional impairments and disability3/10) did not have a line drawn to mental and emotional impact. See lines 759-763. Also Social effects of leprosy (9/10) did not have a line to drawn to Biophysical Impact. See lines 774-776 which talks about treatment adherence.

Reviewer #2: The conclusions are supported by the data presented. The study limitations were clearly stated. The authors discuss how the knowledge about the topic is enhanced by the findings and what further issues are to be studies. The authors indicated how the data can inform public policy and improve care delivery in public health.

Reviewer #3: The authors write their conclusions based on the data presented, especially in the four categories aggregated by them throughout the results, highlighting their usefulness for a greater understanding of professionals and policy makers in relation to the studies included in the review if used in isolation.

The limitations of the study were signaled. In addition, the authors suggest future steps for research, assistance, and public health based on the results obtained in the review.

**Editorial and Data Presentation Modifications?**

Reviewer #1: Check Figure 2 for accuracy based on my comments above.

Reviewer #2: Accept.

Reviewer #3: To further enhance the quality of the study that is being given, I would like to draw attention to a few minor observations that were made throughout the text and propose a revision of these data.

Since the epidemiological record containing data for the year 2020, released in September 2021, is now available, I advise amending reference number nine.

In the results session, topic 3.1, line 259, it is described that 361 articles were excluded. Authors should make it clear to the reader the reason for exclusion at this stage. Also add this data to the PRISMA diagram (figure 1).

On line 376, check the appendix number. According to the data presented, I believe that appendix VI should be written.

On line 403, check the reference made to the illustration in question. I did not find the illustration with identification #2 in the appendix, in the part that refers to the reference study 58.

**Summary and General Comments**

Reviewer #1: This well done qualitative research review demonstrates the importance of using qualitative data to better understand issues leprosy affected persons may face and how these issues impact their physical, social, emotional and economic life. It demonstrates the important voice of including the person affected and their perspective within research studies as well as in monitoring and evaluation activities. The insights obtained can guide priorities in program planning and interventions supported by international, national, local leprosy and NTD programs as well as non-governmental organizations and organizations of persons affected by leprosy and NTDs. 

The greatest limitation was the study had few qualitative studies from India and could only review english studies.

This work could guide future study of qualitative data in non-english studies as well as english. It would be interesting to see if the non-english studies not included in this analysis would find similar or different categories.

Reviewer #2: The study is significant in introducing a systematic approach to perform systematic qualitative literature reviews. It also presented the results in a manner that preserved qualitative aspects of the studies reviewed.

Reviewer #3: The study is very well structured and demonstrates methodological rigor. This conveys reliability regarding the results presented. Although it does not generate new knowledge, it perfectly meets the proposal of the review, which is to add the main evidence already available on the subject. I believe that this study will meet the authors' expectations and may be useful for health professionals and public policy makers in improving the care provided to people living with leprosy.

PLOS authors have the option to publish the peer review history of their article (what does this mean?). If published, this will include your full peer review and any attached files.

Reviewer #1: Yes: Linda Faye Lehman

Reviewer #2: Yes: Zoica Bakirtzief

Reviewer #3: No
---

## [Decision Letter · Decision Letter 1]

24 Aug 2022

Dear Dr Abdul Rahman,

We are pleased to inform you that your manuscript 'Experiences of living with leprosy: A systematic review and qualitative evidence synthesis' has been provisionally accepted for publication in PLOS Neglected Tropical Diseases.

Best regards,

Alberto Novaes Ramos Jr

Academic Editor

Guilherme Werneck

Section Editor

Reviewer's Responses to Questions

**Key Review Criteria Required for Acceptance?**

**Methods**

-Are the objectives of the study clearly articulated with a clear testable hypothesis stated?

-Is the study design appropriate to address the stated objectives?

-Is the population clearly described and appropriate for the hypothesis being tested?

-Is the sample size sufficient to ensure adequate power to address the hypothesis being tested?

-Were correct statistical analysis used to support conclusions?

-Are there concerns about ethical or regulatory requirements being met?

Reviewer #2: The study design is appropriate to address the stated objectives. The study employed well known methods of systematic review, adopting multiple electronic data bases. The strategies of data mining and selection were clearly and coherently described and were appropriately developed.

Reviewer #3: The methodology is clearly described. I have no further considerations on this topic.

**Results**

-Does the analysis presented match the analysis plan?

-Are the results clearly and completely presented?

-Are the figures (Tables, Images) of sufficient quality for clarity?

Reviewer #2: The analysis was well presented and thoroughly described. Clearly it followed the plan and analysis strategy proposed. The results were clearly and thoroughly presented. The figures and tables were very helpful and clearly presented. The data was presented in an ordely an coherent manner.

Reviewer #3: The results are clearly described. I have no further considerations on this topic.

**Conclusions**

-Are the conclusions supported by the data presented?

-Are the limitations of analysis clearly described?

-Do the authors discuss how these data can be helpful to advance our understanding of the topic under study?

-Is public health relevance addressed?

Reviewer #2: The conclusions are supported by the data presented. The study limitations were clearly stated. The authors discuss how the knowledge about the topic is enhanced by the findings and what further issues are to be studied. The authors indicated how the data can inform public policy and improve health care delivery in public health.

Reviewer #3: The conclusion is clear and described based on the results.

**Editorial and Data Presentation Modifications?**

Reviewer #2: Accept.

Reviewer #3: Thank you for considering the suggested changes. I have no further considerations on this topic.

**Summary and General Comments**

Reviewer #2: The paper is very comprehensive, it includes relevant publications from endemic countries and thoroughly discussed the findings in an ordely fashion making it very objective in terms of areas that inform public health policies that are relevant to various contexts. It employs an interesting approach to perform systematic qualitative literature reviews. It also presented the results in a manner that preserved qualitative aspects of the studies reviewed.

Reviewer #3: The authors made the suggested changes, which improved this version. I have no additional suggestions for changes to the article.

PLOS authors have the option to publish the peer review history of their article (what does this mean?). If published, this will include your full peer review and any attached files.

Reviewer #2: **Yes: **Zoica Bakirtzief

Reviewer #3: No

---

## [Editor Report · Acceptance letter]

29 Sep 2022

Dear Dr Abdul Rahman,

We are delighted to inform you that your manuscript, "Experiences of living with leprosy: A systematic review and qualitative evidence synthesis," has been formally accepted for publication in PLOS Neglected Tropical Diseases.

Best regards,

Shaden Kamhawi

co-Editor-in-Chief

Paul Brindley

co-Editor-in-Chief
